# Checklists Are Better Than Reward Models For Aligning Language Models

Vijay Viswanathan $^\heartsuit$   Yanchao Sun$^\clubsuit$  Shuang Ma$^{\clubsuit*}$  Xiang Kong$^\clubsuit$
Meng Cao$^\clubsuit$   Graham Neubig$^\heartsuit$   Tongshuang Wu$^\heartsuit$
$^\heartsuit$ Carnegie Mellon University   $^\clubsuit$Apple

## Abstract

Language models must be adapted to understand and follow user instructions. Reinforcement learning is widely used to facilitate this – typically using fixed criteria such as "helpfulness" and "harmfulness". In our work, we instead propose using flexible, instruction-specific criteria as a means of broadening the impact that reinforcement learning can have in eliciting instruction following. We propose "**Reinforcement Learning from Checklist Feedback**" (**RLCF**). From instructions, we extract checklists and evaluate how well responses satisfy each item—using both AI judges and specialized verifier programs—then combine these scores to compute rewards for RL. We compare RLCF with other alignment methods on top of a strong instruction following model (`Qwen2.5-7B-Instruct`) on five widely-studied benchmarks – **RLCF is the only method to help on every benchmark**, including a 4-point boost in hard satisfaction rate on FollowBench, a 6-point increase on InFoBench, and a 3-point rise in win rate on Arena-Hard. We show that RLCF can also be used off-policy to improve `Llama 3.1 8B Instruct` and `OLMo 2 7B Instruct`. These results establish rubrics as a key tool for improving language models' support of queries that express a multitude of needs. We release our our dataset of rubrics (*WildChecklists*), models, and code to the public.[1]

## 1   Introduction

Language models must follow user instructions to be useful. As the general public integrates language model-based assistants into their completion of daily tasks, there is an expectation that models can faithfully follow the users' requests, which increasingly involve rich, multi-step instructions [Liu et al., 2024a, Zhao et al., 2024, Zheng et al.]. Today's models are almost universally trained to follow instructions via a two-step process: instruction finetuning, followed by reinforcement learning from human feedback (RLHF). Instruction finetuning, where the model is trained to mimic responses generated by annotators [Raffel et al., 2019], has historically been the primary workhorse for imbuing language models with some amount of instruction following ability [Wang et al., 2022, Chung et al., 2022, Xu et al., 2024, Lambert et al., 2024a]. Model developers then frequently employ RLHF, where the model is trained to generate responses that look more like labeled "good" responses than "bad" responses, as a refinement step to decrease the likelihood that the model exhibits predefined poor behaviors (typically harmful behaviors) [Ziegler et al., 2019, Bai et al., 2022]. Unlike "verifiable" tasks where reinforcement learning is a workhorse [DeepSeek-AI et al., 2025, Lambert et al., 2024a, Pyatkin et al., 2025], reinforcement learning remains difficult to utilize for ambiguous or "non-verifiable" tasks, such as instruction following. What would it take to make RL a general-purpose solution at eliciting desirable behaviors in subjective or open-ended settings?

---

$^*$Work performed while at Apple.

[1]Code: `www.github.com/viswavi/RLCF`, Dataset: `www.huggingface.co/datasets/viswavi/rlcf`

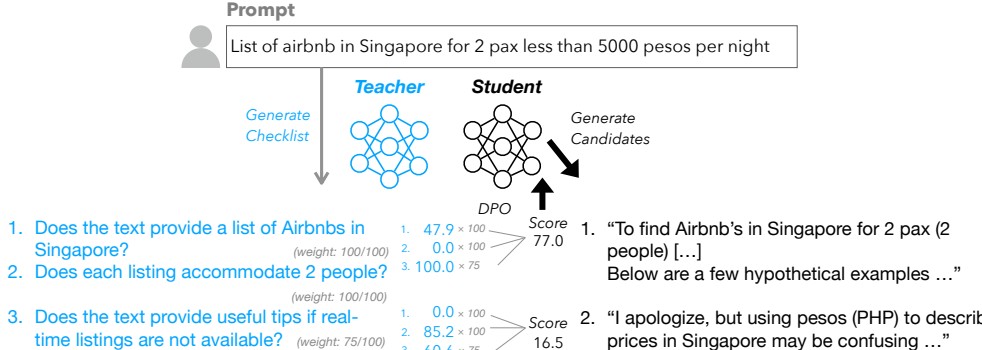

Figure 2: We propose *Reinforcement Learning from Checklist Feedback*, where sampled responses are evaluated by a teacher model grounded on a fixed set of criteria. In our pipeline, given instructions, we first generate checklists synthetically from the instructions, grade each response on each checklist item, combine per-item scores into a single weighted checklist score, then use this score for RL.

We believe the solution must involve producing better reward signals. Recent work on RL for language model alignment has focused on automatically obtaining feedback on model behavior, either by (1) exclusively using *instructions with verifiable answers* [Dong et al., 2024, Pyatkin et al., 2025], (2) grading responses with specially-trained reward models [Wang et al., 2024a, Eisenstein et al., 2023], or (3) distilling preferences from a larger prompted model [Bai et al., 2022, Tunstall et al., 2023]. Using instructions with verifiable answers can only reinforce limited aspects of behavior (ignoring subjective constructs, e.g. topicality or style). Reward models ("RMs") are flexible, but their notion of rewards can be arbitrary, leading to reward hacking [Eisenstein et al., 2023]. When distill-

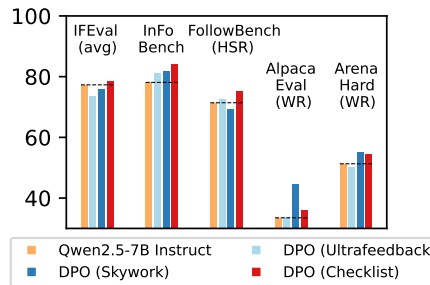

Figure 1: RL on Checklist Feedback consistently improves Qwen2.5-7B-Instruct, whereas every other source of automatic feedback gives mixed results.

ing preferences from a larger model, that model must infer what aspects to grade on, reducing the "generator-verifier gap" that enables RL [Swamy et al., 2025]. Even if multiple criterion-specific prompts are used, these criteria may not be comprehensive [Bai et al., 2022, Glaese et al., 2022].

In this paper, we ask: "how can we grade responses to instructions in a manner that is *automatic* (requires no human annotation), *flexible* (considers all aspects of response quality), *intuitive* (aligned with perceptible differences in responses), and *applicable to any instruction or response*, to enable more better language model alignment?" We propose extracting dynamic rubrics from instructions – an approach we term **Reinforcement Learning from Checklist Feedback (RLCF)**. This approach reduces the task of grading responses to answering a series of yes/no questions, which can be answered by a model or by executing a verification program.

Our key contributions are:

1. We describe a new and improved algorithm for automatically generating checklists at scale.
2. We construct *WildChecklists*, a dataset consisting of 130,000 instructions and corresponding checklists (generated synthetically). When applicable, we accompany items in each checklist with a verification program to facilitate automatic evaluation. We plan to release this dataset to the community as an artifact for future study.
3. We describe a new algorithm for grading responses according to checklists, using language models and code, and we show to use this algorithm to rank responses for preference tuning.
4. We finetune `Qwen2.5-7B-Instruct` via reinforcement learning from checklist feedback using *WildChecklists*, leading to a strong and improved 7B-parameter model for instruction following.

On 5 benchmarks covering both constrained instruction following (IFEval, InFoBench, FollowBench) and general conversational assistance (AlpacaEval, Arena-Hard), we find that RLCF provides benefits on all instruction following benchmarks while maintaining improved performance on general conversational assistance benchmarks. In contrast, all alternative forms of AI feedback lead to mixed results,

as shown in Figure 1. RLCF provides a 5.4% relative improvement over `Qwen2.5-7B-Instruct` in average hard satisfaction rate on FollowBench, a 6.9% relative improvement in overall requirement following ratio on InFoBench, and a 6.4% relative improvement on Arena-Hard [Jiang et al., 2023, Qin et al., 2024, Li et al., 2024]. RLCF can also be used off-policy; we see `Llama 3.1 8B Instruct` and `OLMo 2 7B Instruct` improve using samples collected using `Qwen2.5-7B-Instruct`. Despite these considerable improvements, RLCF simply requires a teacher model, with no need for additional data or human annotations, making this approach amenable to diverse languages or domains. We provide evidence that checklist-based rewards are well-correlated to human preference judgments (comparable to many finetuned reward models) while providing a stronger learning signal than alternatives.

## 2    Checklist Generation

**Desiderata for checklists.** We define a *checklist* as a sequence of *requirements* paired with an instruction that satisfy the following properties:

1. Each requirement in the checklist is a yes/no question (e.g. "Does the text contain 3 commas?").
2. Each requirement in the checklist must be answered relative to a given candidate response.
3. A response would be considered acceptable if and only if the response answers "yes" to all checklist requirements.

To satisfy definition #3, checklists must be *comprehensive* (cover most relevant aspects of quality) and *natural* (entailed by their corresponding instructions). Based on the observation that false positive rewards are often more detrimental to reinforcement learning than false negatives [Huang et al., 2024], we want checklists that are *objective* (facilitate automatic verification) and *atomic* (each requirement focuses on a single aspect of quality), to make requirement checking easier.

**Extract checklists per instruction.** We examine two methods to extract checklists:

- `Direct`: We simply prompt an LM to extract a checklist from a given instruction [Cook et al., 2024]. This approach is intuitive and simple but risks repeating the original instruction via these individual criteria, which may limit *comprehensiveness* and *objectiveness*.
- `Candidate-based`: We view a requirement as any aspect of an instruction that, when absent, causes a response to fail. We propose a two-stage approach: produce responses of varying quality, then prompt an LM to write a checklist of all their possible failure modes. For each checklist item, we also prompt the model to generate an "importance" weight (from 0 to 100).

To compare these, we generate checklists for all instructions in InFoBench [Qin et al., 2024]. We use `gpt-4o` to blindly evaluate each of these checklists on naturalness, objectivity, comprehensiveness, and atomicity, then select the better one overall. We manually perform the same evaluation on a *subset of 50 instructions* from the "Easy Set" of InFoBench.

The results in Table 1 show that checklists generated by prompting an LLM directly are more natural. However, providing candidate responses to the LLM leads to checklists with consistently better objectiveness, atomicity, and overall quality. There are absolute differences between scores from the two evaluations – partly because they use different subsets – but directional trends are consistent. We find that this difference translates to downstream performance after performing RL training. In Section 5.3, we show that Reinforcement Learning from Checklist Feedback is more effective on checklists generated via the candidate-based method.

**Regularization via universal criteria.** In initial experiments, we found that optimizing for checklist completion led to responses beginning with long preamble overviews, suggesting reward hacking. Following Sun et al. [2023] (who report a similar issue in prior work), we add two "universal requirements" to all generated checklists. These requirements state "*1) The response directly address the request without excessive or off-topic information not necessary for addressing the user's instruction? and 2) The response should match the context and the instruction, whether it requires professionalism, friendliness, formality, or neutrality.*", with a corresponding total importance weight of 100/100.

**Dataset Generation** Using the candidate-based method, we generate checklists for 130,000 instructions from WildChat to create a new dataset, *WildChecklists*. To generate candidate responses for our method, we use `Qwen2.5-0.5B`, `Qwen2.5-1.5B`, `Qwen2.5-3B`, and `Qwen2.5-7B` [Yang et al., 2024]. `Qwen2.5-72B-Instruct` is the checklist generator model for both methods.

| Metric | Manual Evaluation | | Automatic Evaluation | |
|---|---|---|---|---|
| | **Direct** | **Candidate-Based** | **Direct** | **Candidate-Based** |
| Naturalness | **94.9** | 93.9 | **88.0** | 85.1 |
| Objectiveness | 88.5 | **91.9** | 88.9 | **89.7** |
| Comprehensiveness | 74.0 | **82.0** | **69.2** | 64.8 |
| Atomicity | 68.0 | **90.0** | 98.6 | **99.0** |
| % Preferred Overall | 38.0 | **56.0** | 40.6 | **51.2** |

Table 1: We evaluate two checklist generation methods on four specific aspects of quality and an overall preference. Manual evaluation is performed on the first 50 rows of InFoBench "easy", while automatic evaluation is performed by `gpt-4o` on all 500 rows of InFoBench.

## 3 Reinforcement Learning from Checklist Feedback

Given *WildChecklists*, we generate high-quality preference data for RL via a four-step process:

**Sampling Candidate Responses.** To facilitate RL, we sample response pairs from our base policy with a temperature of 1.3 and a *top-p* of 0.9 [Holtzman et al., 2019]. This is simpler than prior works that systematically perturb samples to induce greater complexity [Sun et al., 2024, Dong et al., 2024].

**Flexible Scoring** Given a prompt, a response, and an individual checklist item, we use a combination of an LM judge and a verifier program to grade the response. For each checklist item, we prompt a judge model (`Qwen2.5-72B-Instruct`) using the prompt in Appendix B to generate a numerical score between 0 and 100. We take the average of 25 numerical scores sampled from the model.[2].

LLMs struggle to evaluate hard, discrete criteria, such as "does the response contain the letter R at least three times?" [Fu et al., 2024]. To handle such constraints, we follow prior work in generating a program to grade responses when needed [Dong et al., 2024, Zhou et al., 2023]. We prompt a model (using the prompt in Appendix B) to write code only when the model is confident it can exactly check the requirement. For example, our program generator abstains from writing a verifier for the criterion "Is the sentence coherent". The program-graded score is then averaged with the AI judge's score.[3]

**Preference Tuning.** Given a separate numerical score for each criterion for each response, we take the average of these scores, weighted by the importance score of each criterion. We keep only the 40% of response pairs with the greatest difference along at least one criterion of its corresponding checklist. This removes response pairs that are too similar to offer useful pairwise signal. We then assign these responses as a preference pair for direct preference optimization [Rafailov et al., 2023].

## 4 Experimental Setup and Results

### 4.1 Experimental Details

**Training Data** As a fixed source of instructions for all methods, we use WildChat, a set of natural conversations between users and AI language models crowdsourced from users across the world [Zhao et al., 2024]. We filter out conversations that are non-English, toxic, or longer than two turns.

**Models** We experiment with finetuning `Qwen2.5-7B` and `Qwen2.5-7B-Instruct`. To produce AI judgments or ground truth responses, we use `Qwen2.5-72B-Instruct` unless stated otherwise.

**Training** We finetune the model for 2 epochs using DPO with a batch size of 1024 and a maximum sequence length of 2048. We use a cosine learning rate schedule with a max LR of 3e-6 and a min LR of 2e-6.[4] We use OpenRLHF for training [Hu et al., 2024], and we train on one 8xH100 node with 80GB GPU memory, which took roughly 3 hours for each model.

---

[2]We sample responses using the `n` parameter in vLLM [Kwon et al., 2023]. This approach follows prior work that describes the importance of using the mean score rather than mode score from an LM-as-a-judge model [Wang et al., 2025a] Regardless, this makes the AI judge component the computational bottleneck of our pipeline. In Section 5.7, we show that `n` can be significantly reduced, at a modest accuracy cost.

[3]This approach is much simpler than the most relevant prior work that uses programs to evaluate responses, AutoIF [Dong et al., 2024], which uses test-case generation and LM-based filters to remove low-quality programs.

[4]When training models with Ultrafeedback, we instead used a minimum learning rate of 3e-7. We found this parameter resulted in a slightly stronger baseline when learning from this feedback.

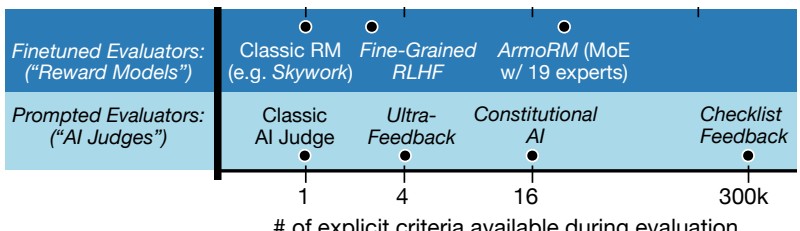

Figure 3: Checklist feedback can be viewed as an extreme mixture-of-evaluators, where the space of (prompted) evaluators is unbounded and a unique subset of evaluators is chosen for each instruction.

**Benchmark Data** We evaluate our method on five benchmarks: IFEval [Zhou et al., 2023], InFoBench [Qin et al., 2024], FollowBench [Jiang et al., 2023], AlpacaEval [Dubois et al., 2024], and Arena-Hard [Li et al., 2024]. The first three of these measure instruction following ability in the presence of fine-grained constraints. The last two measure "general-purpose" instruction following ability, using naturalistic instructions based on user queries collected in the wild.

## 4.2 Baselines

To show that RLCF is more effective than existing approaches, we compare against baselines: *instruction finetuning*, *specially-trained reward models* (using either a single reward or mixture of rewards), and *prompted AI judges* (using either a single evaluation rubric or a mixture of rubrics).

**Instruction Finetuning:** We compare with instruction finetuning, to isolate the benefit of additional knowledge from the manner it is given (ground truth or rewards). Here, we distill [Hinton et al., 2015] from a larger model, `Qwen2.5-72B-Instruct`, finetuned via `LlamaFactory` [Zheng et al., 2024].

**Reward Models:** We mirror our training approach for learning from checklist feedback, but using state-of-the-art reward models to decide which response should be chosen or rejected. We use `Skywork/Skywork-Reward-Gemma-2-27B` [Liu et al., 2024b] and `ArmoRM-Llama3-8B-v0.1` [Wang et al., 2024b] – both are highly rated on RewardBench [Lambert et al., 2024b].[5]

**Prompted AI Judge:** We compare against using the same "teacher" model as a judge, without using rubrics. We query this teacher in two settings: 1) *"Ultrafeedback"*, where the judge rates all candidate responses from 1-5 [Cui et al., 2023] separately across four quality aspects (instruction following, helpfulness, truthfulness, honesty) and averages these scores; and 2) *AI Judge*, where a near-identical prompt as RLCF is used (§3) to similarly sample 25 scores between 0 and 100 from the judge.

In Figure 3, we unify these methods of automatic evaluation to distinguish our method from prior art. In this context, checklist feedback can be viewed as a very large mixture of prompted evaluators.

# 5 Results

## 5.1 RL from Checklist Feedback consistently improves language models

Our proposed approach, RLCF, demonstrates consistent gains across all benchmarks (Table 2, Table 3, and Table 4). On IFEval's "loose" metrics (which apply minor preprocessing to responses before checking for correctness), RLCF improves `Qwen-7B-Instruct` by 2.8-3.0% (relative), as shown in the left half of Table 2. On FollowBench (shown in Table 3), RLCF achieves an 8.2% increase on Constraint Satisfaction Level (CSL; the expected proportion of constraints satisfied) and a 5.5% increase on average Hard Satisfaction Rate (how often all constraints are satisfied). RLCF also performs competitively on InFoBench (right half of Table 2), achieving results comparable to the best-performing reward model-based approaches while maintaining consistent gains across all constrain-based benchmarks. On "general use-case" instruction following benchmarks, RLCF consistently increases the win rate of `Qwen2.5-7B` over `GPT-4` (shown in Table 4), with the relative improvement ranging from 2.8% to 8.4%.

---

[5]`Skywork/Skywork-Reward-Gemma-2-27B` and `ArmoRM-Llama3-8B-v0.1` are ranked as #4 and #24, respectively, on RewardBench as of July 2025.

| | IFEval (prompt) | | IFEval (inst.) | | | InFoBench | | |
|---|---|---|---|---|---|---|---|---|
| | **Loose** | **Strict** | **Loose** | **Strict** | **Avg** | **Easy** | **Hard** | **Overall** |
| GPT-4 | 79.3 | 76.9 | 85.4 | 83.6 | 81.3 | 89.3 | 86.4 | 87.3 |
| *Qwen2.5-7B-Instruct* | 75.0 | 72.5 | 81.8 | 79.9 | 77.3 | 82.7 | 76.0 | 78.1 |
| + SFT (Distilled) | 66.9 | 64.1 | 75.3 | 72.8 | 69.8 | 79.9 | 70.6 | 73.5 |
| + DPO (via Skywork) | 75.8 | 68.0 | 83.2 | 78.5 | 76.0 | 81.0 | 82.4 | 82.0 |
| + DPO (via ArmoRM) | 73.8 | 70.2 | 81.7 | 78.3 | 76.0 | **84.2** | 83.1 | 83.5 |
| + DPO (via Ultrafbk.) | 71.5 | 69.1 | 79.9 | 77.7 | 74.6 | 82.3 | 79.0 | 80.0 |
| + DPO (via AI Judge) | 73.0 | 68.9 | 80.9 | 77.8 | 75.2 | 81.0 | 73.9 | 76.1 |
| + DPO (RLCF) | **77.3** | 72.6 | **84.1** | 80.3 | **78.6** | **84.2** | **84.0** | **84.1** |
| *Qwen2.5-7B (Base)* | 35.7 | 30.5 | 46.6 | 42.1 | 38.7 | 68.8 | 77.4 | 74.8 |
| + SFT on WildChat | 38.1 | 33.5 | 52.2 | 48.6 | 43.1 | 78.1 | 80.1 | 79.5 |
| + DPO (RLCF) | **43.4** | **35.9** | **56.4** | **49.2** | 46.2 | **80.6** | **80.5** | **80.5** |

Table 2: For instruction following, RLCF leads to large gains with open-ended constraints (InFoBench) and slightly positive or neutral changes on format-based constraints (IFEval). Off-the-shelf reward models help on InFoBench but hurt on IFEval. We show positive results (relative to the baseline) in blue, negative in orange, and neutral (within 0.5) in gray; the top variant of a given model is bolded.

| *FollowBench* | Soft Satisfaction Rate | | | | | | Hard Satisfaction Rate | | | | | | |
|---|---|---|---|---|---|---|---|---|---|---|---|---|---|
| **Level** | **L1** | **L2** | **L3** | **L4** | **L5** | **Avg** | **L1** | **L2** | **L3** | **L4** | **L5** | **Avg** | **CSL** |
| GPT-4 | 89.2 | 89.3 | 87.6 | 88.1 | 84.9 | 87.8 | 89.2 | 87.6 | 83.6 | 83.0 | 75.1 | 83.7 | 3.52 |
| *Qwen2.5-7B-Instr.* | 87.4 | 84.0 | 83.0 | 79.6 | 79.0 | 82.6 | 87.4 | 80.6 | 72.3 | 62.2 | 54.4 | 71.4 | 3.05 |
| + SFT (Distilled) | 87.5 | 83.2 | 84.4 | 76.8 | 74.9 | 81.4 | 87.5 | 78.3 | 73.9 | 60.7 | 49.1 | 69.9 | 2.90 |
| + DPO (Skywork) | 79.6 | 84.1 | 77.7 | 77.7 | 78.1 | 79.4 | 79.6 | 81.1 | 67.4 | 62.9 | 56.5 | 69.5 | 2.88 |
| + DPO (ArmoRM) | 86.4 | 84.6 | 79.1 | 79.2 | 76.9 | 81.2 | 86.4 | 82.9 | 69.0 | 63.9 | 49.7 | 70.4 | 3.10 |
| + DPO (Ultrafbk.) | 88.5 | 84.1 | 82.5 | 76.3 | 72.6 | 80.8 | 88.5 | 81.1 | 62.4 | 63.5 | 54.9 | 72.6 | 2.98 |
| + DPO (AI Judge) | 87.2 | 87.9 | 75.7 | 79.2 | 77.6 | 81.5 | 87.2 | 83.5 | 62.4 | 63.5 | 54.9 | 70.3 | 2.95 |
| DPO (RLCF) | **88.6** | **88.8** | 83.8 | 79.9 | 81.0 | 84.4 | **88.6** | 85.2 | 75.8 | 65.1 | 61.8 | 75.3 | **3.30** |
| *Qwen2.5-7B (Base)* | 55.9 | 60.7 | 56.6 | 56.1 | 54.6 | 56.8 | 55.9 | 49.1 | 36.1 | 33.4 | 19.5 | 38.8 | 1.20 |
| + SFT (WildChat) | 65.4 | 75.3 | 71.6 | 64.7 | **65.1** | 68.4 | 65.4 | 69.2 | **57.4** | **46.9** | 40.3 | 55.8 | 2.02 |
| + DPO (RLCF) | 70.6 | 76.0 | 69.5 | 63.6 | 57.8 | 67.5 | 70.6 | 67.7 | 49.6 | 42.4 | 28.3 | 51.7 | 2.08 |
| + RLCF w/o code | **70.9** | **77.1** | **73.3** | **66.0** | 63.5 | **70.2** | **70.9** | **70.0** | 56.5 | 42.9 | **36.3** | 55.3 | **2.20** |

Table 3: RLCF leads to improvements on FollowBench **on across all metrics** when starting with an instruction-tuned model, while using an off-the-shelf reward model for preference labeling leads to regressions for most metrics. This algorithm also helps when applied to a non-instruction-tuned model, though it does not beat supervised finetuning. "CSL" stands for "Constraint Satisfaction Level". We show positive results (relative to the baseline) in blue, negative in orange, and neutral (within 0.5 for satisfaction rate or 0.05 for CSL) in gray; the top variant of a given model is bolded.

## 5.2 Comparing automatic evaluators

In Table 2, Table 3, and Table 4, we observe our approach of performing RL from Checklist Feedback (RLCF) outperforms RL from other sources of automatic evaluation across most benchmarks. However, off-the-shelf reward models show mixed results depending on the benchmark. *Skywork* (`Skywork-Reward-Gemma-2-27B`), a leading model on the RewardBench leaderboard, shows strong improvements with RLHF on InFoBench, Arena-Hard, and AlpacaEval – RLHF via Skywork notably outperforms RLCF on AlpacaEval by a large margin. However, Skywork-guided RLHF leads to notable regressions as on IFEval and FollowBench. Similarly, RLHF with *ArmoRM* shows significant improvements on AlpacaEval and InFoBench, modest/mixed results on Arena-Hard and FollowBench, and significant regressions on IFEval.

We also evaluate checklist feedback's ability as a judge on RewardBench[6]. Table 5 shows checklist scores are well-correlated with preference annotations on RewardBench, especially for the "Chat" and "Chat Hard" categories [Lambert et al., 2024b]. However, specialized reward models (*Skywork*,

---

[6]Unlike our method for checklist generation on WildChat, here we do not use any ground truth or output from other models when generating checklists.

|  | Arena-Hard | | AlpacaEval | |
| --- | --- | --- | --- | --- |
|  | **Vanilla** | **Style-Controlled** | **Vanilla** | **Length-Controlled** |
| GPT-4 (0314) | 50.0 | 50.0 | 22.1 | 35.3 |
| *Qwen2.5-7B-Instruct* | 51.3 | 42.8 | 33.5 | 36.2 |
| + SFT (Distilled) | 32.6 | 29.2 | 36.1 | 33.3 |
| + DPO (via Skywork) | **55.1** | **50.3** | **44.8** | **41.5** |
| + DPO (via ArmoRM) | 50.8 | 46.4 | 37.6 | 38.1 |
| + DPO (via Ultrafeedback) | 52.8 | 47.9 | 33.7 | 38.7 |
| + DPO (via AI Judge) | 51.0 | 44.4 | 28.8 | 33.4 |
| + DPO (RLCF) | 54.6 | 48.4 | 36.2 | 37.1 |
| *Qwen2.5-7B (Base)* | 19.6 | 24.1 | 8.9 | 9.4 |
| + SFT on WildChat | 8.8 | 8.8 | 9.4 | 7.5 |
| + DPO (RLCF) | 19.4 | 21.6 | **11.2** | 10.5 |
| + RLCF w/o program verification | **23.1** | **27.1** | 11.0 | **13.9** |

Table 4: We compare methods on two "general" instruction following benchmarks: Arena-Hard and AlpacaEval. RLCF gives modest but consistent gains on both the original metric and length/style-controlled metric on each benchmark. We show positive results (relative to the baseline) in blue, negative in orange, and neutral (within 0.5) in gray; the top variant of a given model is bolded.

|  | Chat | Chat Hard | Safety | Reasoning |
| --- | --- | --- | --- | --- |
| Skywork-27B | 96.1 | **89.9** | **93.0** | **98.1** |
| ArmoRM | **96.9** | 76.8 | 90.5 | 97.3 |
| Checklist-Based Reward | 90.0 | 80.7 | 71.4 | 88.5 |

Table 5: On RewardBench, Specialized reward models like Skywork-27B and ArmoRM excel at predicting which response is superior. Our checklist-based approach is worse on this this benchmark, but still achieves competitive performance on challenging categories like Chat Hard and Reasoning.

*ArmoRM*), are much better here, though worse at supervising RLHF. This follows prior evidence that reward model "accuracy" is poorly correlated with efficacy in RLHF [Malik et al., 2025, Razin et al., 2025]. Note that our method shows relatively less correlation with Safety – our implementation of RLCF is not designed as a substitute for safety alignment (see more discussion in §Section 5.5).

## 5.3 Learning from candidate-based vs directly-generated checklists

In Section 2, we described a novel method for *candidate-based* checklist generation, and we presented some intrinsic evaluation showing that this method generates good checklists. Do these checklists indeed translate to better models after RL training?

In Table 6, we observe that RLCF is consistently better using "candidate-based" checklists than using checklists generated "directly" by prompting: 2% better on IFEval, equally good on InFoBench, and 2-3% better on FolllowBench. This shows that RLCF depends on detailed, and objective checklists that may offer more new information than checklists obtained directly from the original prompt.

## 5.4 Where does checklist feedback help?

Does checklist feedback help with a specific aspect of instructions, such as rule-based format constraints? Performance on specific constraint types on FollowBench, shown in Table 7, shows that, unsurprisingly, prompt-based scoring is helpful for prompts involving style or format constraints. We also see that **RLCF is best for "content" constraints**, which are qualifiers included on open-ended questions to limit the valid space of answers (e.g. "*How might economic data from the past quarter affect the Fed's decision on interest rates? Additionally, consider how inflation rates might influence their decision.*"). This suggests **checklist feedback incentivizes models to attend to the full instruction** rather than using a few influential spans to generate responses.

This hypothesis is supported by qualitative analysis of the preference data in Table 8. We observe that using an AI judge with a single rubric is often insensitive to major changes in the prompt. When the user asks to translate an utterance to Spanish, the AI judge assigns a 100-point score to both a

|  | IFEval (prompt) | | IFEval (inst.) | | | InFoBench | FollowBench | |
|  | **Loose** | **Strict** | **Loose** | **Strict** | **Avg** | **Overall** | **SSR** | **HSR** |
|---|---|---|---|---|---|---|---|---|
| *Qwen2.5-7B-Instruct* | 75.0 | 72.5 | 81.8 | 79.9 | 77.3 | 78.1 | 82.6 | 71.4 |
| + RLCF (direct) | **74.3** | **69.5** | **81.5** | **77.9** | **76.9** | 84.3 | 82.5 | 72.8 |
| + RLCF (candidate-based) | **77.3** | **72.6** | **84.1** | **80.3** | **78.6** | 84.1 | **84.4** | 75.3 |

Table 6: Using candidate-based checklists is crucial to making RLCF work, suggesting that the quality and properties of checklists are important for learning from checklist feedback.

|  | Avg (HSR) | Format | Style | Situation | Content |
|---|---|---|---|---|---|
| GPT-4 | 83.7 | 83.3 | 97.3 | 78.2 | 76.0 |
| *Qwen2.5-7B-Instruct* | 71.4 | 60.0 | 87.3 | 78.1 | 60.0 |
| + DPO (Skywork) | 69.5 | 62.7 | 88.0 | 74.7 | 52.8 |
| + DPO (ArmoRM) | 70.4 | 62.0 | 89.3 | 71.8 | 58.4 |
| + SFT (Distilled) | 71.1 | 61.3 | 85.3 | 80.0 | 57.6 |
| + RLCF *w/o prompt-based scoring* | 73.6 | 62.7 | 90.7 | **81.8** | 59.2 |
| + RLCF *w/o program verification*) | 73.8 | **68.7** | **91.3** | 80.0 | 55.2 |
| + RLCF | **75.3** | 64.0 | 90.7 | 80.0 | **66.4** |

Table 7: On FollowBench, RLCF helps especially with "content" constraints, which are qualifiers that restrict the valid space of answers. The metric shown is "average hard satisfaction rate". We speculate that RLCF helps models attend to full instructions. We show positive results in blue, negative in orange, and neutral (within 0.5) in gray; the top variant of a given model is bolded.

perfect response and a poor response that contains a slightly flawed translation along with incoherent phrases. In the second example, Skywork-27B assigns wildly different scores to responses with identical meaning. The two scoring components of checklist feedback – a verification program and a checklist-based AI judge – can balance each other's shortcomings, as shown in the first example.

### 5.5 Does RLCF lead to specialization at the expense of generality?

Do the gains shown by RLCF on instruction following come at the expense of domains not well-represented in the training data (WildChat)? WildChat focuses primarily on daily assistance, advice, and analysis (75.5%), with only 12% on factual information and mathematics [Zhao et al., 2024]. Is this causing specialization at the expense of generality? We evaluate RLCF on three tasks with limited representation in WildChat: refusal to answer unsafe prompts (measured on XSTest [Röttger et al., 2023]), basic math (measured on GSM8K [Cobbe et al., 2021]), and hallucination prevention (measured on TruthfulQA [Lin et al., 2021]). In Table 9, we see that RLCF slightly alters the model's safety profile (reducing false refusals while slightly impairing true refusals) and reduces GSM8K and TruthfulQA performance by 1-1.5%. This suggests a need for expanding *WildChecklists* to a more diverse prompt distribution. Fortunately, this is easier than retraining a reward model. RMs like Skywork-27B show better generality out-of-the-box (e.g. in Table 9) because they do not assume fixed criteria, but support for very different prompts would require retraining [Malik et al., 2025].

### 5.6 RLCF improves other model families off-policy

We use `Qwen2.5` models as the policy and judge in our experiments. Does RLCF still work for other model families? Do our checklists capture universal criteria or are they tied to a particular choice of model? Using *WildChecklists* with samples from `Qwen2.5-7B-Instruct` scored by `Qwen2.5-72B-Instruct`, we trained `Llama 3.1 8B Instruct` [Dubey et al., 2024] and `OLMo 2 7B Instruct` [OLMo et al., 2024] off-policy with DPO. In Table 10, we see positive results with both models: `Llama` sees strong improvements with InFoBench/FollowBench after RLCF and `OLMo` sees strong improvements with IFEval after RLCF, and there are no regressions for either model. This suggests these checklists do capture universal criteria.

### 5.7 How much compute is required for producing checklist-based AI judgments?

| Responses | | Skywork Reward | AI Judge | Checklist (code) | Checklist (judge) |
|---|---|---|---|---|---|
| **Prompt** | Translate to Spanish: "Hello how are you doing?" | | | | |
| **Checklist** | 1. Is the generated text in Spanish? (weight: 100/100)
2. Is the text an accurate and complete translation of the English sentence? (weight: 100/100) | | | | |
| "¡Hola, ¿cómo estás?" | | 25.5 | 100.0 | 100.0 | 95.2 |
| "HOLA, ¿CÓMO TE ESTÁScaller"H!impo-rtant"Endpoint unfinished际">vak dao "'圣诞'"[...]" | | 0.0 | 100.0 | 100.0 | 0.0 |
| **Prompt** | make a sentence with "dense" | | | | |
| **Checklist** | 1. Does the generated text contain the word "dense"? (weight: 100/100)
2. Is the generated text a coherent and grammatically correct sentence? (weight: 75/100) | | | | |
| "The forest was dense, with trees so close together [...]" | | 33.1 | 100.0 | 100.0 | 97.3 |
| "The forest was blanketed by a layer of dense vegetation." | | 8.0 | 100.0 | 100.0 | 96.6 |

Table 8: Comparing the scores assigned to various prompts and responses, we see that reward models are too sensitive, prompted AI judges are too granular, and checklists give stable, interpretable scores.

| | XSTest | | | GSM8K | TruthfulQA | |
|---|---|---|---|---|---|---|
| | Safe (↑) | Unsafe (↑) | Overall (↑) | Accuracy | MC1 | MC2 |
| *Qwen2.5-7B-Instruct* | 92.0 | 83.0 | 86.0 | 83.2 | 43.5 | 60.4 |
| + RLCF | **95.6** | 81.0 | 86.9 | 82.2 | 42.0 | 59.0 |
| + DPO (Skywork) | **90.4** | **89.0** | **88.6** | **85.6** | **45.2** | **63.1** |

Table 9: In our experiments, RLCF uses prompts that focus primarily on daily assistance and writing [Zhao et al., 2024]. This has a small effect on "non-target" tasks: refusal to answer unsafe prompts (measured on XSTest), basic mathematical reasoning (measured on GSM8K), and hallucination prevention (measured on TruthfulQA), suggesting that RLCF incurs modest trade-offs on underrepresented domains, which could be mitigated by improving domain coverage of training prompts.

As described in Section 3, RLCF grades responses on each criterion using 25 samples from a judge model. This creates the computational bottleneck of our scoring procedure. In Figure 4, we evaluated models trained using RLCF modified to use fewer samples from the judge. Response grading on WildChat with 3, 5, 10, or 25 samples took 32, 40, 72, and 92 hours, respectively, on one 8xH100 node. We observe stable efficacy on IFEval[7] and InFoBench across sample sizes. For FollowBench, using fewer samples hurts the "content" and "situation" categories, suggesting that a cheap, high-variance score may suffice for simpler criteria but not for difficult, ambiguous instructions.

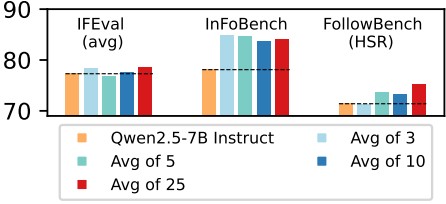

Figure 4: RLCF samples 25 scores when grading each requirement. This is expensive. Fortunately, much of the efficacy is retained using just 5 samples (55% less clock time).

## 6 Related Works

Our method is a new means of generating synthetic AI feedback. This follows prior work that use "AI feedback" to guide reinforcement learning algorithms, either via a single prompt/rubric [Tunstall et al., 2023] or a collection of rubrics [Cui et al., 2023]. In our paper, we show that checklist feedback is significantly more effective than UltraFeedback [Cui et al., 2023], which evaluates responses on four global principles. Our work is also related to prior works that use reward models as synthetic preference annotators for RL [Sun et al., 2023]. In Table 2, Table 3, Table 4, we demonstrates the risks of using reward models to supervise RL [Liu et al., 2024b, Wang et al., 2024c].

---

[7]The models we trained all showed moderate variance on IFEval, so slight differences are likely due to noise.

| | IFEval (prompt) | | IFEval (inst.) | | | InFoBench | FollowBench | |
| | **Loose** | **Strict** | **Loose** | **Strict** | **Avg** | **Overall** | **SSR** | **HSR** |
|---|---|---|---|---|---|---|---|---|
| *Llama 3.1 8B Instruct* | 79.1 | 71.3 | 83.5 | 76.2 | 77.5 | 83.1 | 77.6 | 68.0 |
| + RLCF (off-policy) | **80.2** | **71.9** | 83.9 | 76.5 | **78.1** | **84.2** | **81.8** | **72.3** |
| *OLMo 2 7B Instruct* | 78.1 | 69.3 | 79.5 | 71.3 | 74.6 | 80.1 | 72.3 | 59.4 |
| + RLCF (off-policy) | **79.6** | **70.8** | **82.0** | **73.9** | **76.6** | 80.5 | 71.9 | **60.4** |

Table 10: RLCF works *off-policy* for `OLMo 2 7B Instruct` and `Llama 3.1 8B Instruct`. For `Llama`, we see neutral results with IFEval and strong improvements on InFoBench and FollowBench. For `OLMo`, we see strong improvements with IFEval and neutral results on InFoBench and Follow-Bench.

We focus on complex instruction following. One line of work synthesizes instructions with explicit, pathological constraints to train models to generalize to other instructions [Xu et al., 2023, He et al., 2024, Sun et al., 2024, Dong et al., 2024]. These papers use DPO on controlled candidate responses, while we contribute a drop-in evaluator which allows scoring responses sampled directly from a student model (opening the door to other RL algorithms like GRPO [Shao et al., 2024]).

Our work is related to a nascent line of work that explores using rubrics for language model alignment and evaluation. Cook et al. [2024] demonstrate that using model-generated checklists can be useful at inference-time for proprietary LLMs. Similarly, Saha et al. [2023] use generated checklists at inference time to improve constrained reasoning tasks. [Saad-Falcon et al., 2024] use checklists to evaluate language models and match our finding that checklists can outperform reward models at response evaluation. Wang et al. [2025b], contemporaneously to us, introduce a rubric generation method called "pre-comparison-derived criteria" which also uses candidate responses sampled from different language models; they show that these criteria improve the agreement of automatic evaluations of model-generated responses with human judgments. Our work is the first to apply a similar approach to RL, at the same time as some contemporaneous works. Dineen et al. [2025] use detailed rubrics that greatly improve model safety. Their work differs from ours by using RL with explicit rewards (GRPO) and using checklists that were defined at a global level (the same large set of criteria applied to all tasks) rather than the instruction-specific checklists we espouse in this work.

# 7 Limitations

We highlight three limitations with our work. First, our implementation of RLCF uses "strong-to-weak generalization" – a larger model (`Qwen2.5-72B-Instruct`) provides AI judgments for tuning a smaller model, though RLCF beats other methods also useing a 72B teacher. Second, we only explored preference-based RL in our work. We believe that using checklist feedback to train policy gradient-based algorithms is an exciting future research direction. Lastly, our scoring method is expensive – grading response pairs on each requirement for 130k instructions with `Qwen2.5-72B-Instruct` takes roughly 4 days on eight H100 GPUs with 80GB GPU memory, which is computationally infeasible for many practitioners. In Section 5.7, we show that this cost can be reduced by 50% at some slight cost to accuracy, but further optimization is warranted.

# 8 Conclusion

We provide a detailed study of reinforcement learning from checklist feedback (RLCF). We propose a novel algorithm for automatically extracting rubrics from instructions, and we use this algorithm to construct a dataset of instructions and rubrics, *WildChecklists*. We demonstrate that RLCF is uniformly effective at improving strong instruction following models on all benchmarks we consider.

Our study follows an active line of work that highlights the limitations of reward models in supervising reinforcement learning. One exciting future direction to emerge from this work is: how can we combine checklist-style feedback with trainable judges? Our current approach relies on carefully-designed, prompt-based components for rubric generation and response grading under a rubric. Why is this more effective than methods that naturally learn to grade responses from human preference data? We believe that analysis of RLCF can motivate better reward models in the future.

## Acknowledgements

We thank Saumya Gandhi, Xiang Yue, Gokul Swamy, Apurva Gandhi, Lintang Sutawika, Jessie Mindel, Qianou Ma, Chenyang Yang, and Xinran Zhao for helpful discussions and Akhila Yerukola for invaluable writing assistance and technical advice. Vijay Viswanathan, Graham Neubig, and Tongshuang Wu were supported in part by a grant from Apple. Any views, opinions, findings, and conclusions or recommendations expressed in this material are those of the author(s) and should not be interpreted as reflecting the views, policies, or position, either expressed or implied, of Apple Inc.

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

# A The role of response pair mining

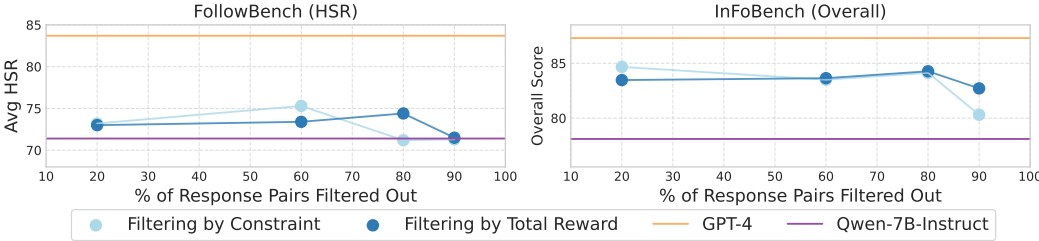

Figure 5: Impact of different filtering strategies on model performance on FollowBench and In-FoBench. We compare filtering pairs based on overall checklist score differences versus filtering based on single-aspect score differences, at varying dataset sizes. There are only slight differences between these two filtering methods, until we start filtering out the vast majority of the data. This suggests that the reward signal, rather than the specific filtering algorithm, is likely responsible for this method's effectiveness.

In our algorithm for learning from checklist feedback, we only train on the 40% of response pairs that differ the most on at least one criterion. This approach differs from thresholding on the reward difference with a single scalar reward, which may represent the aggregation of multiple small differences across all requirements. How much is the filtering component responsible for the success of RLCF?

To investigate this, we compared two approaches: selecting pairs with the largest differences in overall weighted checklist scores versus selecting pairs with the largest differences on any single aspect's score. As shown in Figure 5, performance shows that, when discarding just 20% or 40% of response pairs, the method of filtering makes almost no difference. On the other hand, when discarding 90% of response pairs (with least difference in reward), performance plummets on both benchmarks, suggesting that, regardless of the filtering strategy, keeping some "harder" response pairs is beneficial. Rather than aspect-based filtering being the primary driver of improvement, the results suggest that checklist-based rewards inherently capture more instruction-relevant dimensions of quality, leading to more effective preference tuning even with moderate filtering.

# B Prompt for Generating Verification Programs

We describe the prompt used for generating programs to selectively verify responses in Figure 6.

# C   Prompt for Scoring Semantic Criteria

We describe the prompt used for requirement checking in Figure 7.

## NeurIPS Paper Checklist

1. **Claims**

   Question: Do the main claims made in the abstract and introduction accurately reflect the paper's contributions and scope?

   Answer: [Yes]

   Justification: The main defeasible claim made in the abstract and introduction are that the proposed method is effective on "every" benchmark we evaluate on, and that no other alignment methods share this property. This is supported by the data (see Table 2, Table 3, and Table 4). On one of the 5 benchmarks we try (IFEval), the proposed method has no real effect on 2 out of 4 metrics, but it does help on the other 2 metrics (see Table 2). Otherwise, the positive results are consistent.

   Guidelines:

   - The answer NA means that the abstract and introduction do not include the claims made in the paper.
   - The abstract and/or introduction should clearly state the claims made, including the contributions made in the paper and important assumptions and limitations. A No or NA answer to this question will not be perceived well by the reviewers.
   - The claims made should match theoretical and experimental results, and reflect how much the results can be expected to generalize to other settings.
   - It is fine to include aspirational goals as motivation as long as it is clear that these goals are not attained by the paper.

2. **Limitations**

   Question: Does the paper discuss the limitations of the work performed by the authors?

   Answer: [Yes]

   Justification: Yes, we list a few limitations of the proposed work in Section 7.

   Guidelines:

   - The answer NA means that the paper has no limitation while the answer No means that the paper has limitations, but those are not discussed in the paper.
   - The authors are encouraged to create a separate "Limitations" section in their paper.
   - The paper should point out any strong assumptions and how robust the results are to violations of these assumptions (e.g., independence assumptions, noiseless settings, model well-specification, asymptotic approximations only holding locally). The authors should reflect on how these assumptions might be violated in practice and what the implications would be.
   - The authors should reflect on the scope of the claims made, e.g., if the approach was only tested on a few datasets or with a few runs. In general, empirical results often depend on implicit assumptions, which should be articulated.
   - The authors should reflect on the factors that influence the performance of the approach. For example, a facial recognition algorithm may perform poorly when image resolution is low or images are taken in low lighting. Or a speech-to-text system might not be used reliably to provide closed captions for online lectures because it fails to handle technical jargon.
   - The authors should discuss the computational efficiency of the proposed algorithms and how they scale with dataset size.
   - If applicable, the authors should discuss possible limitations of their approach to address problems of privacy and fairness.
   - While the authors might fear that complete honesty about limitations might be used by reviewers as grounds for rejection, a worse outcome might be that reviewers discover limitations that aren't acknowledged in the paper. The authors should use their best judgment and recognize that individual actions in favor of transparency play an important role in developing norms that preserve the integrity of the community. Reviewers will be specifically instructed to not penalize honesty concerning limitations.

3. **Theory assumptions and proofs**

Question: For each theoretical result, does the paper provide the full set of assumptions and a complete (and correct) proof?

Answer: [NA]

Justification: There is no significant theoretical work contained in this paper.

Guidelines:

- The answer NA means that the paper does not include theoretical results.
- All the theorems, formulas, and proofs in the paper should be numbered and cross-referenced.
- All assumptions should be clearly stated or referenced in the statement of any theorems.
- The proofs can either appear in the main paper or the supplemental material, but if they appear in the supplemental material, the authors are encouraged to provide a short proof sketch to provide intuition.
- Inversely, any informal proof provided in the core of the paper should be complemented by formal proofs provided in appendix or supplemental material.
- Theorems and Lemmas that the proof relies upon should be properly referenced.

4. **Experimental result reproducibility**

Question: Does the paper fully disclose all the information needed to reproduce the main experimental results of the paper to the extent that it affects the main claims and/or conclusions of the paper (regardless of whether the code and data are provided or not)?

Answer: [Yes]

Justification: The paper aims to give full details about the information needed to reproduce the paper. This includes data sources (Section 4.1), training hyperparameters (Section 4.1), exact models used (Section 4.1), data generation parameters (Section 3), and *exact* prompts used (listed in Appendix B and Appendix C).

Guidelines:

- The answer NA means that the paper does not include experiments.
- If the paper includes experiments, a No answer to this question will not be perceived well by the reviewers: Making the paper reproducible is important, regardless of whether the code and data are provided or not.
- If the contribution is a dataset and/or model, the authors should describe the steps taken to make their results reproducible or verifiable.
- Depending on the contribution, reproducibility can be accomplished in various ways. For example, if the contribution is a novel architecture, describing the architecture fully might suffice, or if the contribution is a specific model and empirical evaluation, it may be necessary to either make it possible for others to replicate the model with the same dataset, or provide access to the model. In general. releasing code and data is often one good way to accomplish this, but reproducibility can also be provided via detailed instructions for how to replicate the results, access to a hosted model (e.g., in the case of a large language model), releasing of a model checkpoint, or other means that are appropriate to the research performed.
- While NeurIPS does not require releasing code, the conference does require all submissions to provide some reasonable avenue for reproducibility, which may depend on the nature of the contribution. For example
  (a) If the contribution is primarily a new algorithm, the paper should make it clear how to reproduce that algorithm.
  (b) If the contribution is primarily a new model architecture, the paper should describe the architecture clearly and fully.
  (c) If the contribution is a new model (e.g., a large language model), then there should either be a way to access this model for reproducing the results or a way to reproduce the model (e.g., with an open-source dataset or instructions for how to construct the dataset).
  (d) We recognize that reproducibility may be tricky in some cases, in which case authors are welcome to describe the particular way they provide for reproducibility. In the case of closed-source models, it may be that access to the model is limited in

some way (e.g., to registered users), but it should be possible for other researchers to have some path to reproducing or verifying the results.

5. **Open access to data and code**

Question: Does the paper provide open access to the data and code, with sufficient instructions to faithfully reproduce the main experimental results, as described in supplemental material?

Answer: [Yes]

Justification: The code with careful documentation is posted at `www.github.com/viswavi/RLCF` (along with links to models), and our generated dataset is posted at `www.huggingface.co/datasets/viswavi/rlcf`

Guidelines:

- The answer NA means that paper does not include experiments requiring code.
- Please see the NeurIPS code and data submission guidelines (`https://nips.cc/public/guides/CodeSubmissionPolicy`) for more details.
- While we encourage the release of code and data, we understand that this might not be possible, so "No" is an acceptable answer. Papers cannot be rejected simply for not including code, unless this is central to the contribution (e.g., for a new open-source benchmark).
- The instructions should contain the exact command and environment needed to run to reproduce the results. See the NeurIPS code and data submission guidelines (`https://nips.cc/public/guides/CodeSubmissionPolicy`) for more details.
- The authors should provide instructions on data access and preparation, including how to access the raw data, preprocessed data, intermediate data, and generated data, etc.
- The authors should provide scripts to reproduce all experimental results for the new proposed method and baselines. If only a subset of experiments are reproducible, they should state which ones are omitted from the script and why.
- At submission time, to preserve anonymity, the authors should release anonymized versions (if applicable).
- Providing as much information as possible in supplemental material (appended to the paper) is recommended, but including URLs to data and code is permitted.

6. **Experimental setting/details**

Question: Does the paper specify all the training and test details (e.g., data splits, hyper-parameters, how they were chosen, type of optimizer, etc.) necessary to understand the results?

Answer: [Yes]

Justification: The paper discloses training and test details, such as training data sources (Section 4.1), training hyperparameters (Section 4.1), data generation parameters (Section 3), prompts (listed in Appendix B and Appendix C, and specific benchmark dataset settings Section 4.1.

Guidelines:

- The answer NA means that the paper does not include experiments.
- The experimental setting should be presented in the core of the paper to a level of detail that is necessary to appreciate the results and make sense of them.
- The full details can be provided either with the code, in appendix, or as supplemental material.

7. **Experiment statistical significance**

Question: Does the paper report error bars suitably and correctly defined or other appropriate information about the statistical significance of the experiments?

Answer: [No]

Justification: We did not report statistical significance of our experiments, due to compute overhead and clear trends we see across extensive comparison experiments. This aligns with common practice in the instruction following literature, where statistical testing is often

omitted. We do consider adding more such analysis in future iterations though.and this is something we will consider adding in future drafts.

Guidelines:

- The answer NA means that the paper does not include experiments.
- The authors should answer "Yes" if the results are accompanied by error bars, confidence intervals, or statistical significance tests, at least for the experiments that support the main claims of the paper.
- The factors of variability that the error bars are capturing should be clearly stated (for example, train/test split, initialization, random drawing of some parameter, or overall run with given experimental conditions).
- The method for calculating the error bars should be explained (closed form formula, call to a library function, bootstrap, etc.)
- The assumptions made should be given (e.g., Normally distributed errors).
- It should be clear whether the error bar is the standard deviation or the standard error of the mean.
- It is OK to report 1-sigma error bars, but one should state it. The authors should preferably report a 2-sigma error bar than state that they have a 96% CI, if the hypothesis of Normality of errors is not verified.
- For asymmetric distributions, the authors should be careful not to show in tables or figures symmetric error bars that would yield results that are out of range (e.g. negative error rates).
- If error bars are reported in tables or plots, The authors should explain in the text how they were calculated and reference the corresponding figures or tables in the text.

8. **Experiments compute resources**

Question: For each experiment, does the paper provide sufficient information on the computer resources (type of compute workers, memory, time of execution) needed to reproduce the experiments?

Answer: [Yes]

Justification: For our key experiments, we reported the exact computer resources used (Section 4 for training and Section 7 for data generation). We **did not** report all computer resources for reproducing all baselines or ablation experiments, because we used varying compute for these experiments, and this did not seem absolutely necessary to describe in the paper.

Guidelines:

- The answer NA means that the paper does not include experiments.
- The paper should indicate the type of compute workers CPU or GPU, internal cluster, or cloud provider, including relevant memory and storage.
- The paper should provide the amount of compute required for each of the individual experimental runs as well as estimate the total compute.
- The paper should disclose whether the full research project required more compute than the experiments reported in the paper (e.g., preliminary or failed experiments that didn't make it into the paper).

9. **Code of ethics**

Question: Does the research conducted in the paper conform, in every respect, with the NeurIPS Code of Ethics https://neurips.cc/public/EthicsGuidelines?

Answer: [Yes]

Justification: To the best of our knowledge, we completely conform to the NeurIPS Code of Ethics.

Guidelines:

- The answer NA means that the authors have not reviewed the NeurIPS Code of Ethics.
- If the authors answer No, they should explain the special circumstances that require a deviation from the Code of Ethics.

- The authors should make sure to preserve anonymity (e.g., if there is a special consideration due to laws or regulations in their jurisdiction).

10. **Broader impacts**

    Question: Does the paper discuss both potential positive societal impacts and negative societal impacts of the work performed?

    Answer: [No]

    Justification: The paper does not discuss societal impacts at length, beyond a brief disclaimed in Section 5.2 that our method is not a substitute for safety alignment. We justify this because our paper, which focuses on instruction following, considers the same societal impacts as any other work on instruction following – there is nothing significantly different about the harms posed by our proposed method than those posed by existing, widespread methods (e.g. reward models or AI judges).

    Guidelines:

    - The answer NA means that there is no societal impact of the work performed.
    - If the authors answer NA or No, they should explain why their work has no societal impact or why the paper does not address societal impact.
    - Examples of negative societal impacts include potential malicious or unintended uses (e.g., disinformation, generating fake profiles, surveillance), fairness considerations (e.g., deployment of technologies that could make decisions that unfairly impact specific groups), privacy considerations, and security considerations.
    - The conference expects that many papers will be foundational research and not tied to particular applications, let alone deployments. However, if there is a direct path to any negative applications, the authors should point it out. For example, it is legitimate to point out that an improvement in the quality of generative models could be used to generate deepfakes for disinformation. On the other hand, it is not needed to point out that a generic algorithm for optimizing neural networks could enable people to train models that generate Deepfakes faster.
    - The authors should consider possible harms that could arise when the technology is being used as intended and functioning correctly, harms that could arise when the technology is being used as intended but gives incorrect results, and harms following from (intentional or unintentional) misuse of the technology.
    - If there are negative societal impacts, the authors could also discuss possible mitigation strategies (e.g., gated release of models, providing defenses in addition to attacks, mechanisms for monitoring misuse, mechanisms to monitor how a system learns from feedback over time, improving the efficiency and accessibility of ML).

11. **Safeguards**

    Question: Does the paper describe safeguards that have been put in place for responsible release of data or models that have a high risk for misuse (e.g., pretrained language models, image generators, or scraped datasets)?

    Answer: [No]

    Justification: We have instituted one very limited safeguard, which was to curate our training data only with a filtered, non-toxic version of WildChat. However, as with any work that releases general purpose language models and datasets to support these models, there is a very real potential for misuse. We feel that the risk for misuse is comparable to the many other papers that study language modeling, and we are not targeting sensitive domains specifically in our work.

    Guidelines:

    - The answer NA means that the paper poses no such risks.
    - Released models that have a high risk for misuse or dual-use should be released with necessary safeguards to allow for controlled use of the model, for example by requiring that users adhere to usage guidelines or restrictions to access the model or implementing safety filters.
    - Datasets that have been scraped from the Internet could pose safety risks. The authors should describe how they avoided releasing unsafe images.

- We recognize that providing effective safeguards is challenging, and many papers do not require this, but we encourage authors to take this into account and make a best faith effort.

12. **Licenses for existing assets**

    Question: Are the creators or original owners of assets (e.g., code, data, models), used in the paper, properly credited and are the license and terms of use explicitly mentioned and properly respected?

    Answer: [Yes]

    Justification: We cited and credited our use of WildChat, in compliance of its `ODC-BY` license.

    Guidelines:

    - The answer NA means that the paper does not use existing assets.
    - The authors should cite the original paper that produced the code package or dataset.
    - The authors should state which version of the asset is used and, if possible, include a URL.
    - The name of the license (e.g., CC-BY 4.0) should be included for each asset.
    - For scraped data from a particular source (e.g., website), the copyright and terms of service of that source should be provided.
    - If assets are released, the license, copyright information, and terms of use in the package should be provided. For popular datasets, `paperswithcode.com/datasets` has curated licenses for some datasets. Their licensing guide can help determine the license of a dataset.
    - For existing datasets that are re-packaged, both the original license and the license of the derived asset (if it has changed) should be provided.
    - If this information is not available online, the authors are encouraged to reach out to the asset's creators.

13. **New assets**

    Question: Are new assets introduced in the paper well documented and is the documentation provided alongside the assets?

    Answer: [Yes]

    Justification: As mentioned in the paper, we will release the dataset we constructed *Wild-Checklists*. We will also document and opensource the code for RLCF.

    Guidelines:

    - The answer NA means that the paper does not release new assets.
    - Researchers should communicate the details of the dataset/code/model as part of their submissions via structured templates. This includes details about training, license, limitations, etc.
    - The paper should discuss whether and how consent was obtained from people whose asset is used.
    - At submission time, remember to anonymize your assets (if applicable). You can either create an anonymized URL or include an anonymized zip file.

14. **Crowdsourcing and research with human subjects**

    Question: For crowdsourcing experiments and research with human subjects, does the paper include the full text of instructions given to participants and screenshots, if applicable, as well as details about compensation (if any)?

    Answer: [NA]

    Justification: The paper does not involve crowdsourcing nor research with human subjects. The only annotation work done for this paper was done by the authors of the paper.

    Guidelines:

    - The answer NA means that the paper does not involve crowdsourcing nor research with human subjects.

- Including this information in the supplemental material is fine, but if the main contribution of the paper involves human subjects, then as much detail as possible should be included in the main paper.
- According to the NeurIPS Code of Ethics, workers involved in data collection, curation, or other labor should be paid at least the minimum wage in the country of the data collector.

15. **Institutional review board (IRB) approvals or equivalent for research with human subjects**

Question: Does the paper describe potential risks incurred by study participants, whether such risks were disclosed to the subjects, and whether Institutional Review Board (IRB) approvals (or an equivalent approval/review based on the requirements of your country or institution) were obtained?

Answer: [NA]

Justification: The paper does not involve crowdsourcing nor research with human subjects.

Guidelines:

- The answer NA means that the paper does not involve crowdsourcing nor research with human subjects.
- Depending on the country in which research is conducted, IRB approval (or equivalent) may be required for any human subjects research. If you obtained IRB approval, you should clearly state this in the paper.
- We recognize that the procedures for this may vary significantly between institutions and locations, and we expect authors to adhere to the NeurIPS Code of Ethics and the guidelines for their institution.
- For initial submissions, do not include any information that would break anonymity (if applicable), such as the institution conducting the review.

16. **Declaration of LLM usage**

Question: Does the paper describe the usage of LLMs if it is an important, original, or non-standard component of the core methods in this research? Note that if the LLM is used only for writing, editing, or formatting purposes and does not impact the core methodology, scientific rigorousness, or originality of the research, declaration is not required.

Answer: [Yes]

Justification: We have described using LLMs as teacher and student models, for improving the student model's instruction following capability. We have also detailed the checklist synthesis approach (based on criticizing LLM-generated outputs), as well as using LLM as judges for specific checklist items.

Guidelines:

- The answer NA means that the core method development in this research does not involve LLMs as any important, original, or non-standard components.
- Please refer to our LLM policy (`https://neurips.cc/Conferences/2025/LLM`) for what should or should not be described.

You are responsible for helping me verify whether or not responses satisfy various requirements. Given a natural language requirement, you will have to classify whether this can be converted to a Python program to automatically check it or whether it should be given to a human collaborator. Your human collaborator is a reliable and cheap expert, and you should trust them. Accordingly, only write code for verifying a constraint if you are very confident that this will exactly check the constraint. You should never make ANY approximations when verifying a constraint. If you feel that you must approximate the constraint in order to verify whether a response follows that constraint, let your human collaborator take care of it. You should ONLY generate code for requirements that are explicitly about syntax or format (e.g. punctuation, unicode characters used, number of paragraphs, shallow grammar, presence of some mandatory keyword specified by the prompt, etc). If there are many different ways to write an answer, you most likely should not generate code for it. If you are not sure, you should not generate code. You should only generate code if you are 100% sure that the constraint can be verified perfectly with a simple Python function.

When a constraint can be verified EXACTLY with a program, then return a Python function that verifies the constraint. This code should be contained within two sets of triple backquotes, "`. The Python function must return a boolean, and it should only use builtins/standard libraries in Python. If the constraint cannot be verified with a simple Python function (which means your human collaborator will handle the verification of this constraint), please return "NONE" and nothing else. The safest thing to do is to return "defer to human expert ####" 95% of the time. Now, let's go through a couple examples:

Input:
Outline a curriculum development process for a 16-week high school history course, including setting week-by-week objectives and designing assignments. Include two mid-term exams and a final exam. Provide a detailed grading criteria based on the assignments and exams you have designed.

Requirement:
Does the response specify the inclusion of two mid-term exams and a final exam

Verification Function:
defer to human expert ####
(there are multiple valid ways to describe this, and it is not a simple boolean check)
...

Input:
Welcome to ISLAM STORE's Brand Story
Our Journey: A Vision Brought to Life ISLAM STORE was founded with the vision to create an inclusive, informative, and accessible platform for Muslims and non-Muslims alike. Our goal is to promote awareness and understanding of Islam while offering high-quality Islamic products.

Requirement:
Does the generated text contain any Arabic?

Verification Function:
```python
def verify_requirement(text):
# Arabic Unicode block range (0600-06FF)
# Plus Extended Arabic (0750-077F)
# Plus Arabic Presentation Forms (FB50-FDFF, FE70-FEFF)
return any(('\u0600' <= char <= '\u06FF') or ('\u0750' <= char <= '\u077F') or ('\uFB50' <= char <= '\uFDFF') or ('\uFE70' <= char <= '\uFEFF') for char in text)
```
...

Input:
{input}

Requirement:
{requirement}

Verification Function:

Figure 6: Prompt for generating verification code

Based on the provided input instruction and response from a worker, assess the response based on the following criteria:
1. Does it satisfy the specific requests of the instruction?
2. Does the response directly address the request without excessive or off-topic information not necessary for addressing the user's instruction?
3. Does the response match the context and the instruction, whether it requires professionalism, friendliness, formality, or neutrality?

Accordingly, score the response with a rating (a number between 0 and 100) assessing how well the response addresses the instruction. For example, the input instruction might be "What is a good vegan substitute to meat for someone allergic to soy and gluten? Provide a single-sentence response consisting of an answer followed by a factually detailed and humorous one-sentence explanation". Your selection should be based on the response and the instruction, using the following rating scale:

- 100: Select 100 if the generated text represents an optimal solution that expertly balances all relevant aspects of the instruction. For the example above (about the vegan substitute), and the criterion above (about factual detail), an example 100-point response is "Mushrooms, because they can be easily caramelized and browned, they are rich in the glutamates which lead to incredible umami flavors, they naturally are completely free of soy and gluten, and they don't look cute as babies". This response is richly detailed and factual, and though it fails to be humorous, it is still a 100-point response on the factual detail criterion.
- 75: Return 75 if the generated text very effectively addresses the main requirements but has room for minor improvements. The response should be unconditionally acceptable (at a professional level) but may not be absolutely perfect. There are no mistakes that critically undermine the question. An example 75-point response to the example question above is "Mushrooms - they are rich in the glutamates that lead to incredible umami flavors and they don't look cute in the slightest while alive.". This response has one interesting fact but could be more detailed.
- 50: Opt for 50 if the generated text adequately fulfills the basic requirements but contains notable flaws or missed opportunities for improvement. The response should still be functionally acceptable. The response contains at most one minor inadequacy or inaccuracy related to the question but there are no mistakes that critically undermine the question. An example 50-point response to the example question above is "Mushrooms, because they can be easily caramelized and browned, they're universally beloved by sophisticated palates, and they don't look cute in the slightest while alive." The statement that they're universally beloved by people with sophisticated palates, while potentially true, is vague and not objective.
- 25: Return 25 if the generated text fulfills the key condition specified by the question and demonstrates awareness of the key requirements but fails to execute them effectively. The text may contain non-critical inaccuracies or irrelevant information. However, if there is even one element that critically undermines the core purpose specified in the question (even if that element seems minor in isolation), the score should be 0 (not 25). An example 25-point response to the example question above is "Mushrooms, because they can be easily caramelized and browned, they are absolutely brimming with protein, and they don't look cute in the slightest while alive." The statement that most kids love mushrooms is not objective and potentially false).
- 0: Opt for 0 if the generated text fails to meet the question's requirements or provides no information that could be utilized to answer the question. If the response contains a critical error relevant to the question, return a 0. For the question about the vegan substitute, an example 0-point response is "Mushrooms, because they make you question why you ever thought a dead animal could compare to this vegan delight." While funny and engaging, this response contains zero factual detail about mushrooms, critically violating the question.

Your score can be any number between 0 and 100 (not just the ones listed above). If you are totally confused, return -1 as a default. You should use your judgment to determine the most appropriate score. Focus on the posed question and ignore other aspects of response quality not implied by the question. Return only a number - do not include any other text in your response.

Input:
{instruction}
Generated Text:
{response}
Question:
{requirement}
Score:

Figure 7: Prompt for checklist scoring

