# OpenReview forum: "Checklists Are Better Than Reward Models For Aligning Language Models"
_NeurIPS.cc/2025/Conference — NeurIPS 2025 spotlight_

### Official Review · Reviewer_hvGc · 2025-07-01

**Clarity:** 3
**Significance:** 3
**Originality:** 3
**Rating:** 4
**Confidence:** 4

**Summary:**

This paper introduces "Reinforcement Learning from Checklist Feedback" (RLCF), a novel approach to RL-based alignment by using a checklist as rewards. The method works by automatically generating checklists from instructions that break down complex requirements into a set of smaller yes/no questions, then using both LLM-as-judges and rule-based verification programs to score responses on each checklist item. These scores are combined into weighted rewards for reinforcement learning via Direct Preference Optimization (DPO). The authors create WildChecklists, a dataset of 130,000 instructions with corresponding checklists, and demonstrate that RLCF consistently improves performance across multiple benchmarks.

**Questions:**

- typo in line 194 Feedbac => Feedback
- Is there a compute-efficient way to determine which items in the checklist are more important than the others?

**Ethical Concerns:**

["NO or VERY MINOR ethics concerns only"]

**Limitations:**

yes

**Paper Formatting Concerns:**

- no major concerns

**Quality:**

3

**Strengths And Weaknesses:**

Strengths:
- This paper proposes a novel framework, RLCF, which organizes rewards via a checklist.
- The proposed method shows competitive results across five benchmarks: IFEval, InFoBench, FollowBench, AlpacaEval, and Arena-Hears.
- The method shows consistent improvements where other approaches fail, with particularly strong gains on complex constraint satisfaction tasks (5.4% relative improvement on FollowBench).
- The approach is practically valuable as it requires only a teacher model without additional human annotation.
- The authors provide thorough ablation studies and qualitative analysis.
- The new dataset WildChecklists seems valuable.

Weaknesses:
- In some sense, prior works exist that have already been doing RLCF implicitly (those that check rule-based format checks as well as LLM-as-judges), those no prior work created such a nice name as RLCF. In this regard, the novelty of this paper is more on the conceptualization and more focused empirical analysis than a ground-breaking algorithm development.
- The proposed approach requires repeated calling of a strong reward model (> 70B), thus it can be quite expensive.
- In some sense, it's not surprising that the proposed method would work better than the baselines that were compared, given the denser and higher quality rewards the method provides during RL.

---

> ### Author Rebuttal · Authors · 2025-07-31
>
> We thank the reviewer for their review and careful reading of our paper. We will address their comments step by step:
>
> ```
> In some sense, prior works exist that have already been doing RLCF implicitly (those that check rule-based format checks as well as LLM-as-judges), those no prior work created such a nice name as RLCF. In this regard, the novelty of this paper is more on the conceptualization and more focused empirical analysis than a ground-breaking algorithm development.
> ```
> We thank the reviewer for appreciating our conceptualization, and we agree that prior works exist that apply methods similar to RLCF. However, we would argue that our approach is indeed methodologically novel. Criteria-based feedback has either been used with synthetic instructions (e.g. [UltraIF -- An et al 2025](https://arxiv.org/abs/2502.04153) or with purely code-based evaluation (e.g.  [AutoIF -- Dong et al 2025](https://arxiv.org/abs/2406.13542 or [RLVR – Lambert et al 2024](https://arxiv.org/abs/2411.15124)). The first (criteria with synthetic instructions) does not allow flexible application to custom sets of prompts or arbitrary RL algorithms, as our method does. The latter (verifiable rewards/code-based evaluation) restricts the space of constraints that can be checked. We argue that our proposed method represents a meaningful step forward along this general direction.
>
>
> ```
> The proposed approach requires repeated calling of a strong reward model (> 70B), thus it can be quite expensive.
> ```
> We agree that this can be quite expensive. However, we also compare with repeated calling of the same judge model in the naive “AI judge” baseline, and this significantly underperforms – we therefore claim that our approach’s inference cost is unlikely to be the sole reason for its effectiveness. We also show that this cost can be reduced by a significant factor at a moderate cost to effectiveness. Duplicating our response to reviewer vQmq (originally posted above), we show the impact of the number of samples on policy effectiveness:
>
> |  # of judges | IFEval (Overall Avg) | InFoBench (Overall) | FollowBench (HSR Avg) |
> |---|---|---|---|
> | Qwen-2.5-7B-Instruct   |  77.3 | 78.1 | 71.4 |
> | 3 samples   |  78.3 | **84.8** | 71.3 |
> | 5 samples   |  76.8 | 84.6 | 73.7 |
> | 10 samples  |  77.6 | 83.6 | 73.3 |
> | 25 samples (RLCF)  |  **78.6** | 84.1 | **75.3** |
>
> We can improve on at least 2 out of 3 instruction following benchmarks with as few as 3 calls to the reward model per criterion. We hope that future researchers will build on this approach, in maintaining the benefits of checklist feedback at a lower cost.
>
>
> ```
> In some sense, it's not surprising that the proposed method would work better than the baselines that were compared, given the denser and higher quality rewards the method provides during RL.
> ```
> Given that checklist-based rewards are based on more factors than a sparse binary preference, we agree that this dense reward is likely to help, and this is definitely a key motivation of our approach. However, we can contrast our setting (which has roughly 5 criteria per instruction) with UltraFeedback, which has 4 universal criteria per instruction. Our method's significantly stronger performance demonstrates that criteria with adaptive specificity are more effective than universal criteria – even with comparable density. We also want to discuss the notion of “higher quality” rewards, as this is a rather deep concept – Skywork and ArmoRM exhibit greater agreement with human preferences on RewardBench than Checklist Feedback, but they are worse at supervising RLHF. This suggests that reward quality ought to be measured via both extrinsic metrics (e.g. RLHF) and intrinsic metrics (e.g. RewardBench), suggesting that the quality of checklist feedback is not self-evident. We think this reflects an ongoing evolution in how we, as a community, evaluate reward quality.
>
> ———
>
> **Questions**
>
> ```
> Is there a compute-efficient way to determine which items in the checklist are more important than the others?
> ```
>
> Our current approach, using few-shot prompting to decide the weight, is already relatively compute efficient and seems robust. However, this is an interesting question that we didn’t focus on extensively. A more compute-efficient way to make this determination could include a custom-trained lightweight checklist item classifier, and to use this classifier’s probabilities as weights, but we felt the current approach was sufficiently efficient and accurate to avoid warranting additional complexity.

---

> > ### Author Response · Authors · 2025-08-06
> > **Following Up**
> >
> > Dear Reviewer hvGc, thank you for your initial review! In our rebuttal above, we aimed to provide clarity on the three points you mentioned:
> > 1. novelty
> > 2. cost of repeatedly calling a strong reward model
> > 3. whether this finding is interesting, considering the density of rewards
> >
> > We hope we have been able to add further context around these excellent questions. In particular, perhaps our rebuttal for #2 was incomplete - if you feel that way, let us know and we're happy to provide further clarity.
> >
> > Thank you for your reviewing efforts.

---

### Official Review · Reviewer_oyZA · 2025-07-02

**Clarity:** 3
**Significance:** 3
**Originality:** 3
**Rating:** 5
**Confidence:** 3

**Summary:**

The paper introduces a method to generate rewards during RLHF (which they call "checklists"). Checklists promise to capture a more complete picture of a model response by generating requirements that a good response should fulfil and assigning rewards according to these requirements. A checklist has to satisfy 3 properties (yes, no-question; each requirement must be answered; all requirements have to be satisfied to count as acceptable). It can then be used in a prompt for an LLM-judge to select chosen and rejected datapoints for preference optimisation.

In their empirical investigation, the authors generate checklists using models from the Qwen-family based on a filtered version of WildChat and fine-tune Qwen2.5-7b on their dataset, comparing their checklist-based approach with reasonable baselines, such as reward models, AI judges without checklists and standard SFT. These comparisons demonstrate the advantage of datapoint-specific checklists over other methods.
Additionally, they provide insightful analysis of the results, clearly showing the advantages and fail conditions of their approach as well as of other methods using instructive example cases.

**Questions:**

1. Did you try using smaller teacher models? How detrimental is the downscaling for performance?
2. Instead of sampling 25 times and averaging, you could also use greedy decoding and achieve the same (with less noise), if I am not mistaken? I am curious whether I am missing something here.

**Ethical Concerns:**

["NO or VERY MINOR ethics concerns only"]

**Final Justification:**

I sustain my high evaluation, considering this paper a valuable contribution to the field

**Limitations:**

Yes

**Quality:**

3

**Strengths And Weaknesses:**

### Strengths
The paper has multiple strengths. A high-level summary:
1. _Format_: Besides minor typos, the paper is well-structured and well-written. Graphics, plots and tables sufficiently support the content of the paper.
2. _Topic & solution_: Obtaining precise quality estimations through automatic means is a very relevant topic for practitioners working on optimizing LLM performance. The ability to bootstrap this information from LLMs in a reliable and straightforward way is a exciting prospect. The paper under review makes a promising contribution in that direction.
3. _Insightful empirical results_: The empirical results are insightful, especially the analysis of breaking conditions for different scoring techniques is informative.
4. _Rigour_: The experiments and outcomes are sound, and the work seems to have been conducting with the necessary rigour. The chosen baselines and benchmarks are reasonable

### Weaknesses
There are some minor weakness to the paper under review:
1. Fine-tuned models and models used for data generation are both from the same family. A model from a different model family during fine-tuning would insure generality.
2. Results on safety benchmarks is low. This is unsurprising, as the authors filter out toxic data points. Those datapoints could have, with slight adaptation in the prompt, be useful for improving safety. It is a missed opportunity to see whether the approach is viable beyond instruction following.
3. Online preference optimization (such as PPO and GRPO) are becoming again more prominent in the literature. Besides DPO, the approach seems very promising for online preference optimization as well. An experiment working with e.g. GRPO would have been interesting to see.

### Typos, formatting and so on:
1. Table 2 has a + in front of the Qwen-instruct model.
2. l194: 5.1 heading

---

> ### Author Rebuttal · Authors · 2025-07-31
>
> We thank the reviewer for their positive and careful review. We thank them for appreciating the applicability of our method, and we especially appreciate that they recognized our paper’s rigor and empirical insights, which we are proud of. We hope that we can address any remaining misunderstandings about our method.
>
> ```
> “Fine-tuned models and models used for data generation are both from the same family. A model from a different model family during fine-tuning would insure generality.”
> ```
> *(the following response is a copy of the response to the first concern of Reviewer #1)*
>
> We believe that, due to the measurably objective phrasing of the checklist items we generate, our grading procedure is indeed general and unlikely to depend on the nuances of the student model. To ensure this, during this rebuttal period, we used the preference dataset that we generated for samples from Qwen-2.5-7B-Instruct, and we used this to finetune meta-llama/Llama-3.1-8B-Instruct and allenai/OLMo-2-1124-7B-Instruct via off-policy DPO.
> We show these results for InFoBench, IFEval, and FollowBench below:
>
> **InFoBench/IFEval**:
> |  Model | InfoBench (Overall)  | IFEval (prompt-level strict)  |  IFEval (prompt-level loose) | IFEval (instr-level strict)  |  IFEval (instr-level loose) |
> |---|---|---|---|---|---|
> | Llama-3.1-8B-Instruct   |  83.07 |  71.3  | 79.1  |  76.2 | 83.5 |
> |  + RLCF  |  **84.18** | **71.9** | **80.2**  | 76.5   |83.9|
> | OLMo-2-1124-7B-Instruct  |  80.09 | 69.3  | 78.1 | 71.3  |79.5|
> |  + RLCF |  80.53 | **70.8**  | **79.6**  | **73.9**  |**82.0**|
>
> **FollowBench**:
> |  Model | *Soft* | L1 | L2 | L3 | L4 | L5 |  Avg | *Hard*|  L1 | L2 | L3 | L4 | L5 |  Avg |  CSL |
> |---|---|---|---|---|---|---|---|---|---|---|---|---|---|---|---|
> | Llama-3.1-8B-Instruct ||84.0|81.7|78.0|73.9|70.2|77.6||84.0|77.1|68.2|61.3|49.4|68.0|2.95|
> |  + RLCF ||**85.1**|**86.7**|**80.5**|**78.2**|**78.4**|**81.8**||**85.0**|**83.0**|**72.0**|61.1|**60.2**|**72.3**|**3.08**|
> | OLMo-2-1124-7B-Instruct ||75.9|**75.8**|72.2|**70.7**|67.0|72.3||75.9|69.1|59.2|**49.9**|**43.1**|59.4|2.58|
> |  + RLCF||76.1|74.6|**75.4**|66.5|66.7|71.9||76.1|**70.0**|**66.9**|48.9|40.1|**60.4**|**2.75**|
>
> With Llama, we see moderately positive results with IFEval and very strong improvements with InFoBench and FollowBench. With OLMo, we see very strong improvements with IFEval and neutral results on InFoBench and FollowBench. Considering that both of these are *off-policy*, we expect that the results could be even stronger if trained on samples from these models. *This suggests these checklists do capture universal issues correctly.*
>
> “Results on safety benchmarks is low. This is unsurprising, as the authors filter out toxic data points. Those datapoints could have, with slight adaptation in the prompt, be useful for improving safety. It is a missed opportunity to see whether the approach is viable beyond instruction following.”
>
> Agreement of checklist-based rewards with the safety subset of RewardBench is indeed significantly lower than state-of-the-art, although we should point out that it’s still competitive with many widely used reward models. We observe that, despite our data mixture not including safety-critical prompts, we still maintain minimal degradation in the safety behavior of the trained policy. To this point, we duplicate our response to reviewer vQmq above, where we measure the Qwen model trained with RLCF on XSTest:
>
>
> |  Model | Response (when safe) | Refusal (when safe) | Response (when unsafe) | Refusal (when unsafe) | **Overall F1** |
> |---|---|---|---|---|---|
> | Qwen-2.5-7B-Instruct    |  92.0\% | 8% | 17\%  | 83\%   |86.0|
> |  "" + RLCF  |  **95.6\%** | **4.4%** | 19\%  | 81\%   |86.9|
> |  "" + DPO (Skywork)  |  90.4\% | 9.6\% | **11\%**  | **89\%**   |**88.6**|
>
>
> The reviewer’s point is valid – checklists could very well be useful for evaluating response safety. Our intuition is that response safety (which is often prompt-agnostic – e.g. do not use profane language or demean protected groups) might be best handled as a distinct step or auxiliary reward for promoting the model’s helpfulness. However, we believe the current RLCF method provides space for this as a complementary consideration.
>
>
> “Online preference optimization (such as PPO and GRPO) are becoming again more prominent in the literature. Besides DPO, the approach seems very promising for online preference optimization as well. An experiment working with e.g. GRPO would have been interesting to see.”
>
> We agree that an experiment working with online RL, particularly using policy gradient algorithms, is an exciting opportunity for future work. We intentionally omitted this direction from our paper, as we’ve found that it’s difficult to compare different reward models on their ability to supervise PPO. The appropriate hyperparameters for optimizing a policy using one reward model can vary dramatically from another reward model, so a broad hyperparameter sweep on a validation set would be necessary for such a comparison. Accordingly, following prior work (e.g. https://arxiv.org/abs/2409.13156), we used DPO as a more dependable signal of usefulness for RL, but we absolutely agree that working with GRPO is an exciting continuation to see how far RL from checklist feedback can take us.
>
>
> ———
>
> **Questions**
>
>
> ```
> Did you try using smaller teacher models? How detrimental is the downscaling for performance?
> ```
>
> We did not try using smaller teacher models, as our goal was to achieve peak performance against strong baselines. We believe that this is an interesting avenue to consider for cost and environmental reasons.
>
> ```
> Instead of sampling 25 times and averaging, you could also use greedy decoding and achieve the same (with less noise), if I am not mistaken? I am curious whether I am missing something here.
> ```
>
> Early on in our experiments, we tried this. At that point, we found that while greedy decoding of the judge’s scores was effective when reranking samples from a policy (“best-of-K alignment”), but significantly less effective at supervising DPO. Our theoretical intuition is that sampling and averaging scores enables better consistency of scores, which mitigates reward hacking. Nonetheless, this is an interesting suggestion which we were unable to run in an end-to-end test during the limited rebuttal period.

---

> > ### Comment · Reviewer_oyZA · 2025-08-05
> >
> > I thank the authors for addressing my curiosities.
> > As I did not have any major concern about this paper during the initial review, I maintain my previous (good) evaluation of the submission.

---

### Official Review · Reviewer_Cq9t · 2025-07-03

**Clarity:** 3
**Significance:** 2
**Originality:** 3
**Rating:** 4
**Confidence:** 4

**Summary:**

This paper proposes RLCF (Reinforcement Learning from Checklist Feedback), a method that aligns language models using instruction-specific checklists instead of fixed criteria like helpfulness. By evaluating how well responses meet checklist items, the method provides tailored rewards for training. Applied to Qwen2.5-7B-Instruct, RLCF improves performance across several benchmarks, showing its effectiveness in handling diverse user instructions.

**Questions:**

1. How can we evaluate whether the scores provided by the LLM are well-calibrated?

2. In cases where the instruction is either too difficult or too easy—leading to low discriminability among responses—would RLCF with checklists perform better? The rationale is that providing a checklist allows for more fine-grained evaluation of responses. Exploring this direction could be an interesting avenue for future research.

3. To reduce the variance of the scores from the judge model, the authors mention sampling 25 responses. Could you provide more details on the actual variance observed in practice?

4. Could you elaborate on how you determine whether a verifier program is applicable for a given instruction or checklist item?

5. How critical is the process of generating the weights?

**Ethical Concerns:**

["NO or VERY MINOR ethics concerns only"]

**Final Justification:**

I have carefully read the authors’ rebuttal and appreciate their thoughtful and detailed responses. Several of my concerns were addressed convincingly. However, some key aspects, such as score calibration and the potential variance of judge responses, remain underexplored empirically. Also, the importance of weights is not validated as well. Therefore, I maintain my original rating.

**Limitations:**

See weaknesses

**Paper Formatting Concerns:**

There is no formatting issue

**Quality:**

3

**Strengths And Weaknesses:**

### **Strengths**

1. RLCF is a flexible and intuitive framework that can be applied to any instruction-response pair.
2. The method proposes a novel framework beyond traditional RLHF approaches, such as those relying on trained reward models.

### **Weaknesses**

1. To prevent reward hacking, the method relies heavily on prompt-based techniques, which may be vulnerable to manipulation.
2. The approach depends on the performance of the judge model, which may not be fully accurate and could result in incorrect preference pairs.
3. When using LLM-based judges, score calibration is crucial. However, the paper does not appear to include a calibration procedure for the response scores, nor does it discuss potential issues or mitigation strategies related to this.

---

> ### Author Rebuttal · Authors · 2025-07-31
>
> Thank you for the thought-provoking review, and thank you for appreciating the novelty and flexibility of our proposed approach. We will discuss each of your concerns:
> ```
> To prevent reward hacking, the method relies heavily on prompt-based techniques, which may be vulnerable to manipulation.
> ```
> We would like to reconsider the premise that our method relies heavily on prompt-based techniques. We do use prompts to generate the checklists, grade checklists, and generate code. However, we believe that using multiple prompt-based components in a multi-step process (involving both LM grading and code) should actually reduce vulnerability to manipulation, while also enabling the grading of non-verifiable instructions where programs cannot verify correctness.
>
> ```
> The approach depends on the performance of the judge model, which may not be fully accurate and could result in incorrect preference pairs.
> ```
> We believe that RLCF is fundamentally less dependent on the judge model’s ability than prior methods. By breaking down the judge’s annotation task into measurably objective and atomic yes-or-no questions, we believe the judge’s annotation task here is easier than in traditional AI judges or reward models, which can also be inaccurate. We also add code-based rewards to accommodate situations where the judge is not reliable. Therefore, we argue that while any reward model or judge is imperfect, our proposed architecture has the potential to be significantly more robust to errors by a judge.
>
> ```
> When using LLM-based judges, score calibration is crucial. However, the paper does not appear to include a calibration procedure for the response scores, nor does it discuss potential issues or mitigation strategies related to this.
> ```
> We appreciate this concern about calibration, and this is interesting from a statistical perspective. Score calibration in RL is already taken care of at the level of the learning algorithm, e.g. "variance reduction" techniques (such as using an "advantage" or "value function"). This is already baked into the DPO algorithm, so the key concern for us is the accuracy and distribution of the reward rather than calibration or normalization of the reward.
>
> ————
>
> **Questions**
>
> ```
> How can we evaluate whether the scores provided by the LLM are well-calibrated?
> ```
> As noted above, DPO’s built-in variance reduction mitigates calibration concerns, leaving us to focus on relative reward quality rather than absolute values.
>
> ```
> In cases where the instruction is either too difficult or too easy—leading to low discriminability among responses—would RLCF with checklists perform better? The rationale is that providing a checklist allows for more fine-grained evaluation of responses. Exploring this direction could be an interesting avenue for future research.
> ```
> This is an interesting avenue for future research! Our initial hypothesis was indeed that RLCF is effective in the setting where responses may differ along a specific aspect, rather than across all or multiple possible aspects of response quality. To test this, we made Figure 4 in Appendix A in the paper. This suggests that, instead, RLCF remains effective even when per-criterion differences are small but aggregate quality differences are large, suggesting our method’s robustness may extend further than our original hypothesis. We speculate that better consistency or coverage of the response may be at play.
>
> ```
> To reduce the variance of the scores from the judge model, the authors mention sampling 25 responses. Could you provide more details on the actual variance observed in practice?
> ```
> Due to time restrictions during this rebuttal period, we were not able to rigorously measure the variance of the scores from the judge model – our statement about the variance was based on theoretical notions of ensembling’s effect on variance and bias. For a potential intuition about the impact of our sampling procedure on RLCF’s effectiveness, we duplicate our response to reviewer vQmq above, where we show the impact of the number of judge samples on trained policies:
>
> |  # of judges | IFEval (Overall Avg) | InFoBench (Overall) | FollowBench (HSR Avg) |
> |---|---|---|---|
> | Qwen-2.5-7B-Instruct   |  77.3 | 78.1 | 71.4 |
> | 3 samples   |  78.3 | **84.8** | 71.3 |
> | 5 samples   |  76.8 | 84.6 | 73.7 |
> | 10 samples  |  77.6 | 83.6 | 73.3 |
> | 25 samples (RLCF)  |  **78.6** | 84.1 | **75.3** |
>
> In summary, any number of samples leads to improvements on at least 2 out of 3 instruction following benchmarks, but increasing the number of samples leads to more consistent gains on all of them.
>
>
> ```
> Could you elaborate on how you determine whether a verifier program is applicable for a given instruction or checklist item?
> ```
>
> We make this determination via a prompt, which is described in Figure 5 of the paper. In particular, we include the following phrasing in the prompt: *“You should never make ANY approximations when verifying a constraint [...] You should ONLY generate code for requirements that are explicitly about syntax or format [...] If there are many different ways to write an answer, you most likely should not generate code for it. If you are not sure, you should not generate code.”*. This phrasing leads the code generator module to reduce the frequency that verification programs are produced, minimizing false positives. We chose this conservative approach over more complex filtering methods (e.g., unit tests) to maintain the method's practical simplicity while ensuring high precision. [1]
>
> ```
> How critical is the process of generating the weights?
> ```
> Due to limitations of scope and time, we have not been able to answer this empirically. However, we implemented the weighting of checklist items in one shot, without significant effort spent on tuning this component. The majority of item weights are between 75 and 100, suggesting that most checklist items are being considered. This suggests weighting is a simple and robust component requiring minimal tuning.
>
> [1] Dong et al. 2024. Self-play with Execution Feedback: Improving Instruction-following Capabilities of Large Language Models. Arxiv.

---

### Official Review · Reviewer_vQmq · 2025-07-03

**Clarity:** 4
**Significance:** 4
**Originality:** 3
**Rating:** 5
**Confidence:** 5

**Summary:**

This paper introduces Reinforcement Learning from Checklist Feedback (RLCF), an alignment method to improve how language models follow instructions. RLCF first involves generating a unique checklist and for each user prompt. The checklist consists of yes/no questions that detail the criteria an ideal response should satisfy and an importance weight for each question.

Then, responses from the policy scored against this checklist using a combination of an AI judge and code-based verifiers to generate training pairs for DPO.

The trained policy shows strong performance across 5 instruction following and general quality benchmarks. The authors also release WildChecklists, a new dataset of 130,000 instruction-checklist pairs to aid in future research in this area.

**Questions:**

1. Did you try using fewer than 25 samples for the judgement score?
2. Qualitatively, did you notice other reward hacking patterns like length increase, emojis, verbosity etc.?
3. In table 6, how are the numbers for RLCF w/o prompt-based scoring obtained? Assuming code-based verifiers can be used for far fewer checklist items, it is not clear how a strong reward signal can be obtained.
4. What is the proportion of checklist items having code-based verifiers?
5. Do you have an agreement analysis with the importance weigthing? Or maybe rank correlation numbers?

**Ethical Concerns:**

["NO or VERY MINOR ethics concerns only"]

**Final Justification:**

The additional experiments provided by the authors help address my generalizability concerns and also confirm my hunch that RLCF could lead to minor regressions in areas that it doesn't specifically target. However, this is a smaller issue in my opinions because the tradeoff could be worthwhile compared to training reward model on a specialized task. Through RLCF we can perform more targeted changes to model performance.

**Limitations:**

Yes

**Quality:**

3

**Strengths And Weaknesses:**

### Strengths
 - This paper proposes a novel alignment method that combines dynamic checklist verification and code based verifiers to produce a strong RL signal. While the ideal of checklists for judging LLM responses is not novel, this paper is the first to demonstrate that the signal is useful for RL. It also incorporates a weight parameter for each item on the checklist to properly prioritize more important items over less important ones. This provides a more nuanced signal to the policy.
 - LLM scoring and judgement is great for open-ended nuanced requirements but is not as good for stricter verifiable requirements. The authors recognize this and incorporate generated code-based verifiers to ensure that judgement on these criteria is accurate.
 - The generated checklists are evaluated for quality by humans and a GPT-4o based automatic evaluation system.
 - The paper includes a variety of results across different alignment baselines and evaluation sets demonstrating how generalizable RLCF is.
 - RLCF is much more transparent and interpretable than previous alignment methods, making it easier to discover and fix reward hacking patterns.

### Weaknesses
 - The paper exclusively uses the Qwen family of models for generating the checklist, the base policy, and the AI Judge. While these results are extensive, demonstrating the usefulness of RLCF with WildChecklists, it would be good to show results with a policy from a different family of models like the Llama, Mistral or Gemma models. Since we know from [Panickssery et. al.](https://arxiv.org/abs/2404.13076) that LLMs prefer their own responses, it would be good to verify that the AI judge is not overly strict on responses from other models.
 - Additionally, since the checklist generation is conditioned on responses from a variety of Qwen models, showing alignment results on other models would also help establish that the checklist question capture universal issues correctly.
 - Table 5 shows that the checklist based reward is not as good at providing safety alignment or reasoning alignment signals. This might indicate that models trained using WildChecklists and RLCF also show safety and reasoning losses. Including results on benchmarks like HarmBench and GSM8K would help understand if that is the case.
 - Relatedly, it is also possible that RLCF-trained models hallucinate more or are less truthful. Since the checklists don't emphasize factual accuracy, it is possible that the AI judge misses any such issues. Running evaluations on TruthfulQA could help understand if this is the case.

---

> ### Author Rebuttal · Authors · 2025-07-31
>
> Thank you for the detail-oriented and extremely constructive review, and thank you for appreciating the novelty, generality, and interpretability of our method. We will address your primary concerns point-by-point:
>
> ```
> “The paper exclusively uses the Qwen family of models for generating the checklist, the base policy, and the AI Judge. While these results are extensive, demonstrating the usefulness of RLCF with WildChecklists, it would be good to show results with a policy from a different family of models [...]”
> ```
> We appreciate this perspective that is grounded in recent literature. The “egocentric bias” of LM judges’ towards their own outputs may impede models’ ability to judge samples from another model family (e.g. Llama) [1]. Due to compute constraints, we could not re-generate preference data during the rebuttal period. However, since the judge is responsible for comparing *samples from the same policy* rather than *across different policies*, we believe this potential confounder is not a significant concern.
>
>
> ```
> “Additionally, since the checklist generation is conditioned on responses from a variety of Qwen models, showing alignment results on other models would also help establish that the checklist question capture universal issues correctly.”
> ```
> *(the following response is a copy of the response to the first concern of Reviewer #1)*
>
>
> We believe that the objective phrasing of our checklist items offers a general signal for tuning other models. To test this, during this rebuttal period, we used the preference dataset generated for Qwen-2.5-7B-Instruct to finetune meta-llama/Llama-3.1-8B-Instruct and allenai/OLMo-2-1124-7B-Instruct via off-policy DPO.
> We show results below:
>
> **InFoBench/IFEval**:
> |  Model | InfoBench (Overall)  | IFEval (prompt-level strict)  |  IFEval (prompt-level loose) | IFEval (instr-level strict)  |  IFEval (instr-level loose) |
> |---|---|---|---|---|---|
> | Llama-3.1-8B-Instruct   |  83.07 |  71.3  | 79.1  |  76.2 | 83.5 |
> |  + RLCF  |  **84.18** | **71.9** | **80.2**  | 76.5   |83.9|
> | OLMo-2-1124-7B-Instruct  |  80.09 | 69.3  | 78.1 | 71.3  |79.5|
> |  + RLCF |  80.53 | **70.8**  | **79.6**  | **73.9**  |**82.0**|
>
> **FollowBench**:
> |  Model | *Soft* | L1 | L2 | L3 | L4 | L5 |  Avg | *Hard*|  L1 | L2 | L3 | L4 | L5 |  Avg |  CSL |
> |---|---|---|---|---|---|---|---|---|---|---|---|---|---|---|---|
> | Llama-3.1-8B-Instruct ||84.0|81.7|78.0|73.9|70.2|77.6||84.0|77.1|68.2|61.3|49.4|68.0|2.95|
> |  + RLCF ||**85.1**|**86.7**|**80.5**|**78.2**|**78.4**|**81.8**||**85.0**|**83.0**|**72.0**|61.1|**60.2**|**72.3**|**3.08**|
> | OLMo-2-1124-7B-Instruct ||75.9|**75.8**|72.2|**70.7**|67.0|72.3||75.9|69.1|59.2|**49.9**|**43.1**|59.4|2.58|
> |  + RLCF||76.1|74.6|**75.4**|66.5|66.7|71.9||76.1|**70.0**|**66.9**|48.9|40.1|**60.4**|**2.75**|
>
> With Llama, we see positive results with IFEval and strong improvements with InFoBench and FollowBench. With OLMo, we see strong improvements with IFEval and neutral results on InFoBench and FollowBench. Considering both are *off-policy*, we expect that results could be stronger if trained on samples from these models. *This suggests these checklists do capture universal issues correctly.*
>
>
>  ```
> “Table 5 shows that the checklist based reward is not as good at providing safety alignment or reasoning alignment signals. This might indicate that models trained using WildChecklists and RLCF also show safety and reasoning losses. Including results on [..] HarmBench and GSM8K would help”
> ```
> We thank the reviewer for raising this issue, which we did not originally test. To estimate of RLCF’s effect on safety, we instead ran on XSTest, which directly tests models’ proclivity to refuse to answer if and only if appropriate, in lieu of the more-complicated HarmBench. Below, we show results using GPT-4-as-a-judge to determine whether the model is performing refusal or not.
>
> |  Model | Response (when safe) | Refusal (when safe) | Response (when unsafe) | Refusal (when unsafe) | **Overall F1** |
> |---|---|---|---|---|---|
> | Qwen-2.5-7B-Instruct    |  92.0\% | **8%** | 17\%  | 83\%   |86.0|
> |  "" + RLCF  |  **95.6\%** | 4.4% | 19\%  | 81\%   |86.9|
> |  "" + DPO (Skywork)  |  90.4\% | 9.6\% | **11\%**  | **89\%**   |**88.6**|
>
> These results give some insight into the domain-specific effects of RLCF. RLCF demonstrates a slightly positive change in unweighted F1 on XSNet, with an improvement in the proclivity of a model to answer when safe coupled with a slight increase in unsafe responses. The contrast with Skywork is illustrative – Skywork was explicitly trained on preference data from safety datasets and it naturally excels at safety evaluation but shows relatively poor performance on instruction following tasks.
>
> We can see a similar pattern expressed with mathematical reasoning with GSM8K (8-shot CoT):
>
> |  Model | Exact-Match Accuracy |
> |---|---|
> | Qwen-2.5-7B-Instruct    |  83.2\% |
> |  "" + RLCF  |  82.18\% |
> |  "" + DPO (Skywork)  | 85.6\% |
>
> Here, we see RLCF slightly reduces accuracy on GSM8K, while Skywork (which was also trained in part on math data) improves considerably on this task. This difference actually showcases a key feature of our method: unlike traditional reward models that must be trained on a fixed set of prompts that ideally overlap with the RLHF distribution [2], RLCF allows practitioners to flexibly target specific capability areas by selecting appropriate data sources. WildChat's focus on daily assistance naturally leads to strong improvements in non-verifiable instruction following but limited mathematical reasoning gains. Practitioners interested in other domains can generate checklists for new examples with less effort than required to tune and train a reward model.
>
>
> ```
> “Relatedly, it is also possible that RLCF-trained models hallucinate more or are less truthful. Since the checklists don't emphasize factual accuracy, it is possible that the AI judge misses any such issues. Running evaluations on TruthfulQA could help understand if this is the case.”
> ```
>
> Following the reviewer’s suggestions, we evaluated RLCF on TruthfulQA to understand if RLCF may lead to greater hallucination. We used the multiple-choice evaluation of TruthfulQA, where the model is tasked with determining if an incorrect answer (that is designed to elicit response biases of models) is chosen over the factual answer. “MC1” measures whether the model chooses the true answer among all presented candidates (true or false). “MC2” measures the probability of the true answer, normalized by the total probability of all answers.
>
> |  Model | MC1 | MC2 |
> |---|---|---|
> | Qwen-2.5-7B-Instruct   |  43.5 | 60.4 |
> |  "" + RLCF  |  42.0 | 59.0 |
> |  "" + DPO (Skywork)  | 45.2 | 63.1 |
>
>
> This suggests that RLCF induces a modest decrease in factual accuracy (MC1: -3.4% relative change, MC1: -2.3%). As with math reasoning and safety, this reflects WildChat’s composition: WildChat focuses on daily assistance, advice, and analysis (75.5%), with only  6.1% on factual information and 6.1% on math [3]. This targeted scope explains why RLCF excels at instruction following while offering neutral or slight decreases in other domains. Given the zero-shot nature of Checklist Feedback, incorporating more factuality-focused prompts during RL training would likely address this limitation, and this is something that the flexibility of checklist feedback allows.
>
> Note: For the above three experiments, we evaluate using OpenInstruct [4].
>
>
> [1] Koo et al. 2024. Benchmarking Cognitive Biases in Large Language Models as Evaluators. In Findings of ACL 2024.
> [2] Malik et al 2025. RewardBench 2: Advancing Reward Model Evaluation. arXiv.
> [3 Zhao et al 2024. WildChat: 1M ChatGPT Interaction Logs in the Wild. In ICLR 2024.
> [4] Wang et al. 2023. How Far Can Camels Go? Exploring the State of Instruction Tuning on Open Resources. In NeurIPS 2023 Datasets and Benchmarks.
>
> ————
>
> **Questions**
>
> ```
> Fewer than 25 samples for the judgement score?
> ```
> Yes, and we will include these results in our final draft.
>
> *(the following section is duplicated in our responses to reviewers Cq9t and hvGc)*
> |  # of judges | IFEval (Overall Avg) | InFoBench (Overall) | FollowBench (HSR Avg) |
> |---|---|---|---|
> | Qwen-2.5-7B-Instruct   |  77.3 | 78.1 | 71.4 |
> | 3 samples   |  78.3 | **84.8** | 71.3 |
> | 5 samples   |  76.8 | 84.6 | 73.7 |
> | 10 samples  |  77.6 | 83.6 | 73.3 |
> | 25 samples (RLCF)  |  **78.6** | 84.1 | **75.3** |
>
> Much of the efficacy is retained using just 5 samples (taking 55% less data generation time). Given the reusability of this data for other model families shown above, we think this justifies this one-time computational investment.
>
>
> ```
> Qualitatively, did you notice other reward hacking patterns like length increase, emojis, verbosity etc.?
> ```
> We observe a 7% increase in word count for responses to IFEval (from 195 to 210). We occasionally see unsolicited rationales (around 5% of time -- this suggests reward hacking, but this is also seen after RL with other reward models. Importantly, RLCF improves on both length-controlled and non-length-controlled metrics on AlpacaEval – length increases, but so does estimated utility.
>
> ```
> In table 6, how are the numbers for RLCF w/o prompt-based scoring obtained? Assuming code-based verifiers can be used for far fewer checklist items, it is not clear how a strong reward signal can be obtained.
> ```
> We only retain prompts with >=1 associated verification program, and rewards are using the verifiable subset. While imperfect, this mirrors the standard setup for RL from verifiable rewards.
>
> ```
> What is the proportion of checklist items having code-based verifiers?
> ```
> 166182 out of 637770 total checklist items (26.1\%) have code-based verifiers.
>
> ```
> Do you have an agreement analysis with the importance weighting? Or maybe rank correlation numbers?
> ```
> Unfortunately, we currently lack ground truth checklist weights to measure this. This is an interesting step for future work.

---

> > ### Author Response · Authors · 2025-08-06
> > **Following Up**
> >
> > Dear Reviewer vQmq, thank you for your efforts as a reviewer! in our response to your initial review, posted a couple days back, we tried to offer further context on a couple points you brought up:
> >
> > 1. exclusive use of the Qwen model family as both response generator and judge
> > 2. exclusive use of the Qwen model family as the policy being aligned
> > 3. analysis of negative interference for safety and reasoning alignment
> > 4. analysis of negative interference on hallucination
> > 5. improving computational efficiency of rubric grading
> >
> > Your suggestions were excellent, and we believe the paper is significantly more well-rounded as a result of it. If you'd like any further clarification on any of these points or new concerns, don't hesitate to let us know. Thank you.

---

> > > ### Comment · Reviewer_vQmq · 2025-08-07
> > > **Response to the authors**
> > >
> > > I thank the authors for the additional results showing the generalizability of  WildChecklists and RLCF. I also appreciate the additional insight that through RLCF we can affect more targeted alignment to models during post-training. While this can lead to small regressions in other areas, it could be a worthwhile tradeoff compared to training a new reward model on a specialized task. I have updated my score to reflect this.

---

### Comment · Area_Chair_EuGt · 2025-08-05
**Please respond to the authors' rebuttal**

Dear reviewers,

Can I request you to take a look at the authors' rebuttal and see if it has addressed the issues raised in your review? Currently there are no responses to the authors' rebuttal.

It is important to acknowledge the rebuttal and give feedback, even if you believe that the issues were not addressed. I would appreciate if you could respond to the authors at the earliest (the discussion period ends in a few days).

Thank you for your time in reviewing the submissions and providing feedback.

Best,
Your AC

---

### Note · Authors · 2025-08-16

Dear Area Chairs and PC's,

We appreciate the reviewers' excellent reviews of our paper. We would like to review the strengths of our work as described by reviewers and summarize our primary responses to their concerns.

The reviewers highlighted the following **strengths of our paper**:
1. **Novel and flexible algorithm:** RLCF provides a flexible algorithm for judging any instruction-response pair, going beyond traditional RLHF and needing only a teacher LM without human annotation or trained reward models (Reviewers Cq9t, oyZA, hvGc)
2. **Very strong empirical results:** Consistent improvements across multiple widely-studied benchmarks where other state-of-the-art alignment methods struggle (Reviewers vQmq, oyZA, hvGc)
3. **Methodological rigor:** we took care to include strong baselines, multiple ablations, and analysis of failures, and we appreciate that reviewers recognized this (Reviewer oyZA)

The reviewers discussed a few key concerns. We believe that these concerns are either naturally overcome by our method or by appropriate usage of it, and we will provide the described evidence of this in our final draft:
1. **Model family restrictions:** Reviewers vQmq and oyZA mentioned our exclusive use of Qwen family models. In our rebuttal, we showed substantial boosts on Llama-3.1-8B and OLMo-2-1124-7B via off-policy DPO, showing that checklists are general and transferable.
2. **Domain coverage:** Reviewer vQmq mentioned the limited discussion of safety, hallucination, and reasoning alignment in our draft.  In follow-up experiments, we found that RLCF does not lead to major regressions on other tasks, such as XSTest, GSM8K, and TruthfulQA. However, RLCF's performance reflects the composition of our training data (WildChat's focus on instruction following and life assistance), and this offers freedom for model developers.
3. **Computational cost:** Reviewers vQmq and hvGc mentioned that RLCF requires repeated calls to strong reward models. We found that we can make our data generation procedure significantly more efficient (by 55-88%) while preserving most of the efficacy of RLCF.

*We will include a discussion of each of these supporting experiments in our final revision.*  We appreciate the reviewers' concerns and believe that, in light of our responses, none invalidate the merit and potential interest in our paper.

We are grateful for the detailed and positive review process and appreciate your efforts and service.

---

### Decision · Program_Chairs · 2025-09-17

**Decision:**

Accept (spotlight)

**Comment:**

The paper presents a new way to obtain reward for RL finetuning, by creating a checklist of properties that a response should satisfy.

Strengths: an innovative idea that can work in situations where a verifiable reward is hard to obtain, flexible method that can be easily plugged for training, strong empirical results
Weaknesses: Only qwen family of models used as both model and judge

The authors provided results on training a llama model during rebuttal, and also some experiments on safety dataset, which have alleviated the main weakness of the paper. I agree with the reviewers that this is a useful and timely contribution to the RL finetuning literature, and suggest the authors to include the rebuttal experiments in the main paper.